# AUHead: Realistic Emotional Talking Head Generation via Action Units Control

**Jiayi Lyu**[1], **Leigang Qu**[2], **Wenjing Zhang**[1], **Hanyu Jiang**[1,4], **Kai Liu**[3],
**Zhenglin Zhou**[3], **Xiaobo Xia**[2], **Jian Xue**[1,*], **Tat-Seng Chua**[2]
[1]University of the Chinese Academy of Sciences [2]National University of Singapore
[3]Zhejiang University [4] State Key Laboratory of Communication Content Cognition,
People's Daily Online *Corresponding author
{lyujiayi21,zhangwenjing242,jianghanyu231}@mails.ucas.ac.cn
{leigangqu,xiaoboxia.uni}@gmail.com, dcscts@nus.edu.sg
{kail,zhenglinzhou}@zju.edu.cn, xuejian@ucas.ac.cn

## Abstract

Realistic talking-head video generation is critical for virtual avatars, film production, and interactive systems. Current methods struggle with nuanced emotional expressions due to the lack of fine-grained emotion control. To address this issue, we introduce a novel two-stage method (**AUHead**) to disentangle fine-grained emotion control, *i.e.*, Action Units (AUs), from audio and achieve controllable generation. In the first stage, we explore the AU generation abilities of large audio-language models (ALMs), by spatial-temporal AU tokenization and an "emotion-then-AU" chain-of-thought mechanism. It aims to disentangle AUs from raw speech, effectively capturing subtle emotional cues. In the second stage, we propose an AU-driven controllable diffusion model that synthesizes realistic talking-head videos conditioned on AU sequences. Specifically, we first map the AU sequences into the structured 2D facial representation to enhance spatial fidelity, and then model the AU-vision interaction within cross-attention modules. To achieve flexible AU-quality trade-off control, we introduce an AU disentanglement guidance strategy during inference, further refining the emotional expressiveness and identity consistency of the generated videos. Results on benchmark datasets demonstrate that our approach achieves competitive performance in emotional realism, accurate lip synchronization, and visual coherence, significantly surpassing existing techniques. Our implementation is available at https://github.com/laura990501/AUHead_ICLR

## 1 Introduction

Audio-driven talking head generation focuses on creating realistic facial animations given the driving audio (Zhou et al., 2025; Zhang et al., 2024). It aims to achieve accurate synchronization, preserve speaker identity, and express vivid and emotionally coherent facial movements (Wang et al., 2021; 2024). This task has gained widespread attention due to its potential applications in virtual avatars (Ma et al., 2025; Zhou et al., 2025; Wu et al., 2022), film production (Gao et al., 2025; Wang et al., 2022; Liu et al., 2025b), and interactive agents (Guan et al., 2023; Cui et al., 2024b).

Existing methods typically feed input audio and a target portrait into a generative model directly (Fig. 1(a)), achieving strong performance in lip synchronization (Chung & Zisserman, 2017; Xu et al., 2024) and identity preservation (Wang et al., 2025). However, they often fall short in producing natural and subtle expressions, as they overlook the deeper emotional cues embedded in speech. To generate videos that are not only temporally accurate but also rich in *emotional expres-*

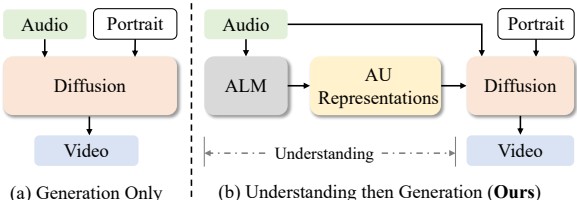

(a) Generation Only    (b) Understanding then Generation (**Ours**)

Figure 1: **Framework comparison between existing talking head generation and our AUHead.** (a) Direct generation from audio and portrait. (b) Our method: audio understanding via ALM, and then generation.

*siveness* and *fine-grained facial dynamics*,
we adopt the framework shown in Fig. 1(b): We leverage an Audio Language Model (ALM (Chu et al., 2023)) to reasonably incorporate world knowledge and to uncover the deep emotional and relational patterns in audio (Qu et al., 2025a;b;c). The high-level cues are subsequently transformed into explicit representations that condition the generation stage.

To achieve this goal, we seek a structured intermediate representation that bridges the audio modality and the visual modality derived from a reference image. **Facial Action Units (AUs)** (Ekman & Friesen, 1978), which describe localized muscle movements with semantic meaning, offer a compact and interpretable expression space. Unlike coarse-grained emotion labels, AUs offer abundant spatial and intensity information, the combinations of which can represent a broad spectrum of facial expressions (Gan et al., 2022; Yan et al., 2019). Since speech often involves coordinated facial muscle movements and carries rich emotional information, AUs are naturally suited to capture both articulatory and emotional cues. We therefore propose to *disentangle AU representations from audio* to guide the facial generation process in a *controllable* manner.

Despite these advantages, leveraging AUs as intermediate control signals introduces two challenges: (i) Due to limited audio-AU paired data and the sparse and subtle nature of AU activations, accurately and efficiently disentangling AU sequences from audio signals is the first challenge. (ii) AU-guided generation is non-trivial due to the controllability versus quality dilemma. Thus, effectively integrating AU features into video synthesis models without compromising lip synchronization, speaker identity, and visual quality becomes the second challenge.

In this work, to address the above challenges, we present a two-stage method, named **AUHead**, for generating realistic and emotional videos conditioned on a reference image and audio. First, we propose to disentangle AU sequences from audio based on an ALM (Chu et al., 2023). Specifically, we propose a **spatial-temporal AU tokenization** strategy to efficiently discretize and tokenize raw dense AU vectors into a compact set. And then we present a **coarse-to-fine AU generation** method inspired by Chain-of-Thought (CoT) (Wei et al., 2022). Lightweight fine-tuning of the ALM can stimulate the potential of emotion understanding and expression during the pre-training phase, by capturing subtle emotional cues like pitch and rhythm, demonstrating its *emotional intelligence* beyond simple speech recognition. In the second stage, we propose an **AU-driven controllable generation** framework, including *AU representation, context-aware AU embedding, and AU-vision interaction*. Specifically, a diffusion-based model takes AU representations, the original audio, and the portrait image as conditions to produce emotional and realistic talking-head videos. Furthermore, to achieve flexible and accurate AU control, we introduce a **disentanglement guidance** strategy specifically tailored for AU conditioning, enabling us to balance AU control with auxiliary conditions (*e.g.*, audio cues and motion priors) and visual quality. Quantitative and qualitative experiments demonstrate that our method outperforms baseline approaches, achieving smoother, more detailed, and emotionally consistent animations. **The primary contributions of this paper include:**

- We propose a novel two-stage method for emotional talking-head generation that decouples Emotion-oriented speech understanding from video synthesis.

- We propose to excavate the emotion understanding and AU generation abilities of ALMs by disentangling emotion-aware AU sequences from raw audio. To our best knowledge, we are the first to explore generating facial AU sequences through ALMs. This establishes AUs as a meaningful and generalizable control space for audio-driven facial animation.

- We present a flexible AU-driven controllable generation framework to generate emotional and realistic talking head videos by AU representation, context-aware embedding, and AU-vision interaction. To strive for a better balance between AU and other conditions, we propose an AU-based disentanglement guidance strategy during inference.

## 2  RELATED WORK

### 2.1  TALKING HEAD GENERATION

The rapid evolution of multi-modal priors(Qu et al., 2025a;b;c) has catalyzed significant breakthroughs across diverse generative frontiers, (Shen et al., 2025a; 2024; Shen & Tang, 2024). Early talking head generation methods convert speech into facial motion with lip-sync discrimination,

head-pose prediction, and compact pose/emotion codes. Wav2Lip adds a lip-sync discriminator (Prajwal et al., 2020). Audio2Head predicts 6D head pose and dense motion from audio (Wang et al., 2021). PC-AVS learns a compact pose code (Zhou et al., 2021). EAMM injects emotion offsets in the same latent space (Ji et al., 2022). Diffusion models (Jin et al., 2025) have become the dominant paradigm for audio-conditioned portrait videos, scaling to long duration, high resolution, and richer controls. DiffTalk, Diffused Heads, EMO, and Loopy generate from audio with latent or autoregressive diffusion (Shen et al., 2023; Stypułkowski et al., 2024; Tian et al., 2024; Ferdowsifard et al., 2021). SadTalker (Zhang et al., 2023) generates 3D motion coefficients and implicitly modulates a 3D-aware face renderer. Sonic (Ji et al., 2025) focuses on global audio perception, disentangling it into intra-clip and inter-clip audio knowledge. DAWN and IF-MDM enable non-autoregressive sampling or compressed motion latents (Cheng et al., 2024; Yang et al., 2024). Hallo, Hallo2, and Long-Term TalkingFace factorize lip, expression, and pose or introduce motion priors for long-form and 4K synthesis (Xu et al., 2024; Cui et al., 2024a; Shen et al., 2025b). Ani-Portrait, HunyuanPortrait, MODA, and PortraitTalk adopt two-stage or controllable pipelines (Wei et al., 2024; Xu et al., 2025; Liu et al., 2023; Nazarieh et al., 2024). EDTalk (Tan et al., 2024a) introduced an efficient disentanglement framework that separately models mouth shape, head pose, and emotional expression through lightweight, orthogonal latent spaces. EchoMimic and V-Express guide diffusion with sparse facial landmarks (Chen et al., 2024; Wang et al., 2024). LatentSync and EchoMimicV2 add alignment and conditioning modules (Li et al., 2024; Meng et al., 2025). EMO-Portraits reshapes the latent space to capture intense and asymmetric expressions (Drobyshev et al., 2024). Recent works have explored leveraging high-level semantic cues from text or multimodal inputs. Wan-S2V (Gao et al., 2025) builds upon a base model to achieve enhanced expressiveness for film-level cinematic contexts. MultiTalk (Kong et al., 2025) addresses multi-person conversational video generation using a Label Rotary Position Embedding (L-RoPE) to solve audio-person binding. OmniHuman-1.5 (Jiang et al., 2025) utilizes Multimodal Large Language Models to synthesize structured textual guidance, enabling semantically coherent and expressive character animations beyond simple rhythmic synchronization. Emotion adapters such as ETAU(Lyu et al., 2024; 2025), EAT(Gan et al., 2023) and SAAS(Tan et al., 2024b) attach a compact affect code to a frozen generator , while MEMO(Zheng et al., 2024) couples a memory-guided module with emotion-aware diffusion, sustaining identity and lip-sync across clips. DICE-Talk (Tan et al., 2025) introduces a disentangled emotion embedder and correlation-enhanced emotion conditioning for emotional portraits. Takin-ADA (Lin et al., 2024) uses a two-stage approach with specialized loss and advanced audio processing to enhance subtle expression transfer and lip-sync precision. In contrast to prior methods that rely on emotion labels or latent codes, our method AUHead, guides a diffusion model through temporally aligned AU features. This enables structured and interpretable control over fine-grained facial expression synthesis.

## 2.2 LLMs for Audio Information Perception

Instruction-tuned Audio-Language Models (ALMs) endow large language models with a "sense of hearing", enabling open-ended spoken interaction across speech, music, and ambient sounds (Tang et al., 2024; Liu et al., 2025a). Qwen-Audio scales multi-task pre-training to over 30 tasks and introduces hierarchical tags to reduce cross-task label interference, establishing a strong open-source baseline (Chu et al., 2023). WavLLM adopts dual encoders plus curriculum learning to improve robustness on universal speech benchmarks (Hu et al., 2024), while SpeechVerse freezes foundation models and instruction-tunes a lightweight fusion layer, outperforming task-specific baselines on eleven benchmarks (Das et al., 2024). While existing ALMs focus on general audio understanding, our method adapts them for a novel purpose: exploring the capacity of ALMs to understand and represent emotional cues through temporally aligned AU sequences.

## 3 Method

**Preliminaries.** Latent Diffusion Models (LDMs) (Rombach et al., 2022) operate by encoding images into a latent space, modeling the latent distribution, and then decoding back to the image space. The encoding-decoding procedure is achieved by a variational auto-encoder (VAE) (Kingma et al., 2013), denoted as $\mathcal{D}(\mathcal{E}(\cdot))$. The model $\epsilon_\theta$ learns to predict the injected noise in the latent space by minimizing:

$$\mathcal{L} = \mathbb{E}_{I, c, t, \epsilon \sim \mathcal{N}(0, I)}\left[\left\|\epsilon - \epsilon_\theta(z_t, t, c)\right\|_2^2\right], \tag{1}$$

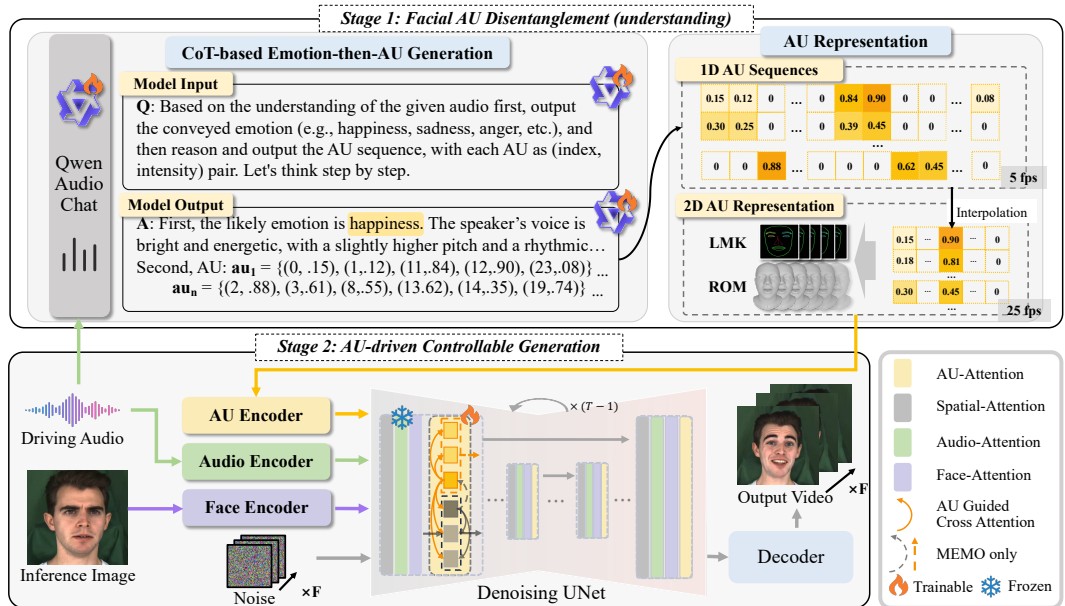

Figure 2: **Overview of the two-stage AU-guided talking head generation framework.** Stage 1 stimulates the AU generation abilities of audio-language models to get 24-dimensional AU sequences from input audio, capturing facial motion dynamics. Stage 2 models the interaction between AU and visual facial representations in a diffusion model to synthesize identity-preserving, emotionally expressive, and lip-synchronized facial animations.

where $\boldsymbol{I}$ denotes an image example, $\boldsymbol{c}$ is a conditioning input (*e.g.*, text, images, or other modality prompts), $\boldsymbol{\epsilon}$ is the sampled Gaussian noise, and $t$ represents the diffusion timestep (with $t \sim \mathrm{Unif}\{0, \ldots, T\}$). The noised latent $\boldsymbol{z}_t$ is obtained from $\boldsymbol{z}_0 = \mathcal{E}(\boldsymbol{I})$ by the forward diffusion process under a predefined schedule (Ho et al., 2020). During inference, we sample $\boldsymbol{z}_T \sim \mathcal{N}(\boldsymbol{0}, \boldsymbol{I})$ and iteratively apply the learned reverse process conditioned on $\boldsymbol{c}$ to obtain a clean latent $\boldsymbol{z}_0$, which is then decoded into the final image via $\mathcal{D}(\boldsymbol{z}_0)$.

## 3.1 FRAMEWORK OVERVIEW

We present a two-stage framework to generate emotional and realistic facial animations, conditioned on speech audio and a reference portrait image, as illustrated in Fig. 2.

● **Stage 1**. We elicit the *emotion understanding and expression* abilities of an audio language model (ALM), *i.e.*, Audio-Qwen-Chat (Chu et al., 2023) to disentangle AU information from audio, by fine-tuning it with AU sequences as supervision. This enables the ALM to generate audio-aligned AU sequences during inference.

● **Stage 2**. We introduce an AU-driven controllable generation framework that enhances *emotion rendering* of a diffusion model via AU representation, AU embedding, and an AU-vision cross-attention mechanism. Compared to emotion labels, explicit AU sequences serve as more fine-grained controllable signals to drive emotional facial animations.

## 3.2 STAGE 1: FACIAL AU DISENTANGLEMENT FROM AUDIO

This stage focuses on disentangling AU sequences from speech signals, which involves (1) understanding the relations between speech acoustics and facial dynamics, *i.e.*, imagining vivid facial expressions solely from audio, and (2) AU generative modeling. Toward this end, we propose to excavate the potential of emotion understanding and AU generation capabilities of a pretrained ALM. In detail, this task requires the ALM to generate a sequence of AU activation vectors at a fixed rate[1] over $T'$ timesteps:

$$\mathbf{AU}_{1:T'} = [\mathbf{au}_1, \mathbf{au}_2, \ldots, \mathbf{au}_{T'}], \quad \mathbf{au}_t \in \mathbb{R}^n. \tag{2}$$

---

[1]In this work, we set the rate to 5 AU sequences per second.

Each vector $\mathbf{au}_t$ (with $n = 24$ in our implementation), conveys a semantically meaningful representation of facial muscle activity, with each dimension corresponding to the intensity of a specific facial muscle group (*e.g.*, lips, jaw, cheeks, and eyebrows). As a fine-grained signal for conveying emotion, the AU sequence $\mathbf{AU}_{1:T'}$ offers strong interpretability, completeness, and compactness.

**Spatial-Temporal AU Tokenization.** In this work, we propose to directly generate AU sequences by viewing them as natural language. Inspired by prior work that converts continuous signals into tokens (Borsos et al., 2023; Wang et al., 2023), we tokenize AU vectors rather than predict them directly with regression. Note that this design not only (1) *aligns with the native language modeling task*, but also (2) *makes use of the abundant knowledge acquired during pre-training*.

However, directly generating AU sequences is non-trivial due to their high density. On average, a 4-second video at 25 FPS yields an AU sequence of approximately 13K tokens, posing significant challenges for the context window and AU modeling capacity of the ALM. To address this issue, we propose a *spatial-temporal tokenization* scheme for AU sequences.

For the spatial aspect, the observation of the inherent sparsity of AU activations (*i.e.*, statistically only $7/24$ AUs are activated) motivates us to tokenize the raw dense AU vector $\mathbf{au}_t$ into a compact set of index–intensity pairs:

$$\hat{\mathbf{au}}_t = \{(i, \mathbf{au}_{t,i}) | \mathbf{au}_{t,i} > \lambda\}, \tag{3}$$

where $\lambda$ denotes a sparsity coefficient. Note that a larger $\lambda$ encourages a sparser transformation, potentially at the cost of losing subtle facial expression details. This strategy[2] substantially reduces the output sequence length[3], simplifies prediction, and facilitates alignment with the discrete token format expected by the ALM. Although the number of active AUs per frame varies, the fixed index order preserves consistency, and the ALM is naturally capable of processing variable-length token sequences. Furthermore, to balance temporal resolution and modeling complexity, we downsample the AU supervision sequence uniformly by a factor $\gamma$. A smaller $\gamma$ results in lower temporal resolution. This temporal compression preserves essential facial expression dynamics while significantly reducing computational overhead. Importantly, the raw audio is always processed at its original sampling rate; only the AU targets are temporally downsampled. Since generating AUs at 25 fps would exceed the ALM's context length, we adopt 5 fps as a practical compromise to keep the sequence tractable. This preserves the main key dynamics, while short-term losses are partly compensated by the continuity of AU trajectories and the generative model.

**CoT-based Emotion-then-AU Generation.** Based on the AU tokenization scheme, we transform dense AU vectors into discrete tokens compatible with language models, enabling direct generation of AU sequences by modeling voice-to-face dependencies. To further exploit the potential of AU modeling with an ALM, inspired by the success of CoT for multistep reasoning (Wei et al., 2022), we propose a coarse-to-fine generation strategy. Specifically, leveraging the correlation between emotional states and AU activation patterns, the ALM *first predicts an emotion category* (*e.g.*, happiness, sadness, and anger) from the audio input, *and subsequently generates the corresponding AU sequence*. Similar to CoT, this coarse-to-fine AU approach enhances AU disentanglement from audio by explicitly incorporating the intermediate emotion prediction as high-level context into the autoregressive decoding process. After generation, we convert $\hat{\mathbf{au}}_t$ back to its dense form $\mathbf{au}_t$ for the subsequent AU representation and embedding.

### 3.3 STAGE 2: AU-DRIVEN CONTROLLABLE GENERATION

While audio-driven diffusion models have shown impressive performance in synthesizing lip-synchronized talking heads, they often generate videos with neutral facial expressions, struggling with conveying rich emotional nuance and fine-grained facial detail, as illustrated in Fig. 1. In this stage, we address this limitation by *explicitly controlling facial expressions using AU sequences* generated in Stage 1. AU sequences enable fine-grained expression control for generating emotionally expressive facial animations, ensuring consistent identity, accurate audio-lip synchronization, and temporally coherent facial movements.

---

[2]For instance, a dense AU vector `[0.38, 0.45, 0.0, ..., 0.0, 0.84, 0.90, 0.0]` is tokenized into a sparse set `{(0,.38), (1,.45), (21,.84), (22,.90)}`.

[3]The sequence length can be reduced by 80.95% on average.

To this end, we propose a controllable generation framework for synthesizing talking-head videos conditioned on an AU sequence, an audio waveform, and a reference image. To enable faithful AU-driven control in a pre-trained diffusion model, we introduce three key components: AU representation, context-aware AU embedding, and AU-vision interaction.

**AU Representation.** To align with the target frame rate, we apply linear interpolation to upsample the generated low-resolution AU sequence in Stage 1 by the factor $1/\gamma$. To further enhance expression control during emotion rendering, we design an AU-to-2D mapping that projects each AU into a 2D facial representation. We consider two options for this representation: keypoint-based Landmark (LMK) (Liu et al., 2024; 2022) and Rendering-of-Mesh (RoM) (Liu et al., 2022). This mapping *transforms 1D AU sequences into a structured and spatially interpretable format*, providing explicit facial topology as informative guidance for generation.

**Context-Aware AU Embedding.** To enhance temporal coherence in facial expressions, we design a context-aware embedding for AU representations. For the $t$-th target frame, we construct a local temporal window of length $n$ to capture local context for modeling continuous expression dynamics. Specifically, we concatenate the AU features within this window and encode them using a lightweight temporal convolutional network, yielding the AU embedding:

$$c_t = \text{Conv}_{\text{AU}}\left([\mathbf{au}_{t-n}, ..., \mathbf{au}_t, ..., \mathbf{au}_{t+n}]\right), \tag{4}$$

where $\mathbf{au}_t$ denotes the upsampled AU representation at the $t$-th frame, and $c_t \in \mathbb{R}^D$ is the resulting AU embedding, with $D$ as the embedding dimension. We collect the AU embedding corresponding to each frame and get $c^{\text{AU}} = [c_1, \ldots, c_{\hat{T}}]$. This approach captures short-term expression patterns and promotes temporal smoothness. By integrating both historical and future AU cues, it enables fine-grained control over subtle and continuous facial movements critical for realistic animation.

**AU-Vision Interaction.** To achieve AU-driven controllable talking-head generation, we facilitate cross-modal interaction between context-aware AU embeddings and visual latents. Specifically, we inserted an *AU adapter comprising multiple cross-attention layers* into the backbone of the pre-trained diffusion model, with zero-initialization for stable training. At each diffusion timestep $t$ and spatial resolution $s$, the intermediate visual latent $z_t^{(s)}$ attends to the AU embeddings $c^{\text{AU}}$ via cross-attention:

$$\hat{z}_t^{(s)} \leftarrow \text{CrossAttn}\left(z_t^{(s)}, c^{\text{AU}}\right). \tag{5}$$

This cross-modal interaction mechanism enables the model to adaptively refine visual facial latents with AU-guided expression cues at every denoising step.

## 3.4 Training and Inference

In Stage 1, given speech as input, we fine-tune the VLM using LoRA (Hu et al., 2022), with ground-truth AU sequences as supervision. Following the language modeling paradigm (Brown et al., 2020), we train the model via next-token prediction using cross-entropy loss. At inference time, the model generates AU sequences conditioned on the input speech.

In Stage 2, we train the AU adapter while keeping all other components frozen, using the diffusion loss defined in Eqn. (1), where $c$ comprises three conditions: audio, inference image, and AU embedding. To support unconditional modeling, each condition is randomly zeroed during training. Inspired by prior guidance strategies (Rombach et al., 2022; Yi et al., 2025), during inference we introduce a *disentanglement guidance* strategy specifically tailored for AU conditioning. Specifically, we can separately modulate the corresponding strengths through adjustable guidance scales. The overall guidance process is formulated as:

$$\hat{\epsilon} = \mathcal{L}_\theta(z_t, \phi, c^{\text{AU}}) + s^H \cdot \left[\mathcal{L}_\theta(z_t, c^H, \phi) - \mathcal{L}_\theta(z_t, \phi, \phi)\right] + s^{\text{AU}} \cdot \left[\mathcal{L}_\theta(z_t, c^H, c^{\text{AU}}) - \mathcal{L}_\theta(z_t, c^H, \phi)\right], \tag{6}$$

where $\phi$ denotes the null condition, and $s^{\text{AU}}$ and $s^H$ are guidance scales controlling the influence of AU and other conditions, respectively.

## 4 EXPERIMENTS

### 4.1 EXPERIMENT SETUP

**Datasets.** We conducted experiments on two widely used emotional talking face datasets: MEAD (Wang et al., 2020) and CREMA (Cao et al., 2014). The MEAD dataset consists of 10,000 high-quality talking face video clips across eight emotion categories. We followed the standard identity-based train/test splits adopted in EAMM (Wang et al., 2020) and EAT (Gan et al., 2023) to ensure subject-independent evaluation. The CREMA dataset contains 7,442 video clips from 91 actors performing six emotional expressions with varying intensity levels. We followed the official subject-based splits provided by Bigioi et al. (Bigioi et al., 2024) and DAWN (Cheng et al., 2025) to guarantee consistent evaluation protocols. To standardize the data, all videos were uniformly resampled to 25 frames per second and resized to 512×512 pixels. The corresponding audio was resampled to 16 kHz. For audio-visual synchronization, we extracted mel-spectrograms using a window size and hop length of 640 samples.

**Implementation Details.** In Stage 1, we trained Qwen-Audio-Chat on 4×NVIDIA A100 GPUs for approximately 24 GPU-hours with a learning rate of $1 \times 10^{-4}$. The sparsity coefficient $\lambda$ and the downsampling factor $\gamma$ were set to 0 and 0.2, respectively. In Stage 2, we adopted Hallo V1 (Xu et al., 2024) and MEMO (Zheng et al., 2024) as base models. Both models were trained on 4×NVIDIA A100 GPUs for 12 GPU-hours. The learning rates were $5 \times 10^{-6}$ for Hallo V1 and $1 \times 10^{-5}$ for MEMO. The window size for context-aware AU embedding is 5 ($n = 2$ in Eqn. (4)). During inference, both Stage-1 AU prediction and Stage-2 AU-driven generation were performed on a single NVIDIA A100 GPU.

**Evaluation Metrics.** We employed multiple quantitative metrics for comprehensive evaluation. Specifically, PSNR, SSIM, and FID (Heusel et al., 2017) were used to assess visual quality, structural similarity, and perceptual realism, respectively. For audio-visual synchronization, we used SyncNet scores (Chung & Zisserman, 2017) and Mouth Landmark Distance (M-LMD) to evaluate lip-sync accuracy. Facial Landmark Distance (F-LMD) was applied to verify the preservation of facial structure and head pose. We further evaluated emotional expression accuracy $\text{ACC}_{\text{emo}}$ using the emotion classification model (Gan et al., 2023). Additionally, Precision, Recall, and Mean Absolute Error (MAE) were used to measure AU regression accuracy in Stage 1 and to verify whether the generated videos in Stage 2 correctly followed the AU guidance.

### 4.2 ABLATION STUDIES

**Stage 1: Ablation Study on CoT-based Coarse-to-Fine AU Generation.** We conducted an ablation study to evaluate the effectiveness of the CoT strategy in Stage 1, as shown in Table 1. Four input-output configurations were tested: (1) Audio and emotion as input to predict AU sequences. (2) Audio only as input

Table 1: Performance of different input-output combinations on AU and emotion prediction. A: Audio input; E: Emotion label; AU: Action Unit sequence.

| Input | Output | Recall | Precision | Accuracy | F1 | MAE | ACC$_{\text{emo}}$% |
|---|---|---|---|---|---|---|---|
| A+E | AU | 0.74 | 0.72 | 0.61 | 0.71 | 0.1928 | – |
| A | AU | 0.63 | 0.65 | 0.50 | 0.62 | 0.2447 | – |
| A | AU→E | 0.66 | 0.68 | 0.53 | 0.65 | 0.2200 | 51.76 |
| A | E → AU | **0.71** | **0.71** | **0.58** | **0.69** | **0.2085** | **67.01** |

to predict AU sequences. (3) Audio as input to predict AUs first, followed by emotion. (4) Audio as input to sequentially predict emotion first, followed by AUs (CoT strategy). Configuration (4), which follows our proposed CoT strategy, significantly outperformed the others in AU regression accuracy. The model achieved 67% accuracy in emotion classification, and 71% precision and recall for AU predictions. For AU intensity regression, the Mean Absolute Error (MAE) was 0.2, which is comparable to the inter-annotator variability reported in FEAFA (Yan et al., 2019; Gan et al., 2022). This indicates that our predictions are within the level of human annotation consistency. The results show that introducing emotion as an intermediate step improves AU estimation and helps the model capture semantic information. This highlights the benefit of coarse-to-fine hierarchical AU reasoning. We also note that generating valid dense AU vectors remains challenging, largely due to the limited context window of Qwen-Audio-Chat and the redundancy in dense AU representations, further motivating our sparse tokenization design.

**Stage 2: Ablation Study in Video Generation Stage.** As shown in Fig. 3, the effect of AU classifier-free guidance (AU CFG scale) on FID, emotion accuracy, and AU regression (MAE) is evaluated. As the scale increases, emotion accuracy improves and MAE decreases, indicating better expression control through AU features. Emotion accuracy and MAE show similar trends, reflecting consistent expression quality. FID first decreases then increases, consistent with previous findings

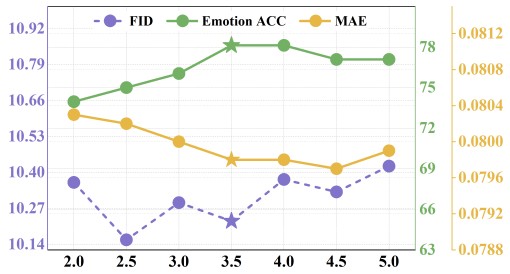

Figure 3: Impact of different AU guidance scales on visual quality (FID) and emotion expression (Emotion ACC and MAE). ★ : the best quality-emotion trade-off.

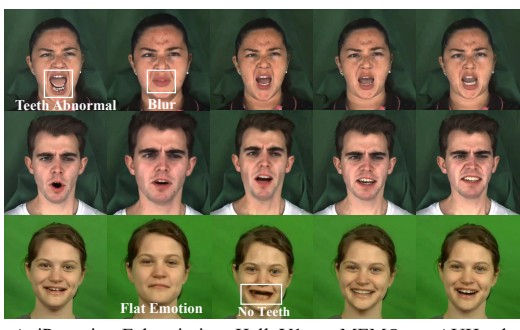

AniPortrait    Echomimic    HalloV1    MEMO    AUHead

Figure 4: Qualitative comparison with SOTA methods on MEAD and CREMA.

Table 2: Ablation results on different AU representations for video generation. The top-2 results are marked in **bold** (best) and underlined (2nd-best). * indicates results reproduced under the same training data and settings for fair comparison. AU Seq: 1D AU sequence; LMK: 2D keypoint-based landmark; RoM: 2D rendering of mesh.

| Dataset | MEAD | | | | | CREMA | | | | |
|---|---|---|---|---|---|---|---|---|---|---|
| Metrics
Method | Sync (↑) | PSNR (↑) | SSIM (↑) | FID (↓) | M/F-LMD (↓) | Sync (↑) | PSNR (↑) | SSIM (↑) | FID (↓) | M/F-LMD (↓) |
| Baseline (MEMO*) | **6.9885** | 23.1910 | 0.7345 | 11.1237 | 2.0684/2.2473 | 6.0922 | 24.2808 | 0.7410 | 8.3881 | 1.9678/2.4296 |
| MEMO + AU Seq | 6.7445 | 23.1666 | 0.7322 | 11.1105 | 1.9060/2.2097 | **6.2857** | 24.2713 | 0.7394 | 8.4159 | 1.9525/2.4257 |
| MEMO + LMK | 6.6311 | 23.3466 | 0.7395 | 10.9671 | 1.8608/2.1604 | 6.2050 | 24.2912 | 0.7413 | **8.2361** | **1.9313/2.3991** |
| MEMO + RoM | 6.6095 | **23.3585** | **0.7399** | **10.8701** | **1.8602/2.1536** | 6.1833 | **24.3113** | **0.7417** | 8.3352 | 1.9339/2.4025 |

like Stable Diffusion (Rombach et al., 2022). The best trade-off occurs at scale 3.5, which balances visual quality and expression accuracy.

To study the impact of different AU representations, we conducted ablation experiments on MEAD and CREMA with identical training settings. Results are shown in Table 2. Note that the baseline MEMO model is retrained on our data split to ensure a consistent comparison. "MEMO + AU Seq" uses a 1D AU sequence as input, whereas "MEMO + LMK" and "MEMO + RoM" adopt 2D AU representations via keypoint landmarks (LMK) and mesh renderings (RoM). This design allows us to compare the effectiveness of spatial versus non-spatial AU encodings under the same framework. From the results, it can be observed that while the model can learn from AU features in all forms, the 1D AU sequence provides weaker conditioning for the diffusion model, leading to suboptimal performance. In contrast, the 2D representations provide stronger spatial priors, resulting in significant improvements across PSNR, SSIM, FID, and LMD metrics on both datasets. Interestingly, the Sync score slightly drops when using 2D AU inputs. One possible reason is that the additional spatial information may increase the model's focus on expression accuracy, slightly compromising the temporal alignment between lip movements and audio. Despite this, overall generation quality and expression consistency are notably improved with 2D AU inputs.

### 4.3 COMPARISON WITH STATE-OF-THE-ARTS

**Quantitative Comparison** We present a quantitative comparison against state-of-the-art methods on the MEAD and CREMA datasets, considering only those that use the same input setting (see Table 3). Overall, AUHead achieves strong performance across most metrics. In particular, we achieve higher PSNR and SSIM scores, indicating improved visual fidelity and structural coherence, as well as lower FID, demonstrating that AU-based expression modeling enhances the perceptual realism of generated videos. In addition, AUHead achieves lower M-LMD and F-LMD scores, indicating more accurate lip geometry and improved facial structure preservation. These results suggest that AU conditioning enhances the fidelity of mouth movements and expression details. The slight drop in Sync confidence may be caused by timing mismatches between the predicted AUs and the speech signal. However, human evaluation (*cf.*, Section 4.3.1) indicates no noticeable audio-lip misalignment, suggesting that this limitation has minimal impact in real-world applications.

Table 3: Comparison of state-of-the-art audio-driven talking head generation methods on MEAD and CREMA benchmarks. The best results are marked in **bold** (best) and underlined (2nd-best). M/F-LMD denotes mouth/face landmark distance.* indicates results reproduced under the same training data and settings for fair comparison.

| Dataset | MEAD | | | | | CREMA | | | | |
|---|---|---|---|---|---|---|---|---|---|---|
| Method \ Metrics | Sync (↑) | PSNR (↑) | SSIM (↑) | FID (↓) | M/F-LMD (↓) | Sync (↑) | PSNR (↑) | SSIM (↑) | FID (↓) | M/F-LMD (↓) |
| Wav2lip (2020) | **8.7778** | 23.0296 | **0.7395** | 32.8043 | 2.6386/2.3885 | 6.7109 | 24.2081 | **0.7533** | 25.6218 | 2.3794/2.4439 |
| Audio2Head (2021) | 6.7809 | 19.6335 | 0.6056 | 35.1387 | 3.3227/3.5371 | 5.7673 | 21.1881 | 0.6533 | 25.0426 | 2.4033/3.2905 |
| Sadtalker (2023) | 7.0015 | 20.9015 | 0.6660 | 28.7729 | 2.8840/2.8841 | 5.3062 | 22.0340 | 0.6814 | 23.6910 | 2.3035/2.8571 |
| V-Express (2024) | 7.2952 | 15.9265 | 0.6023 | 32.2410 | 2.5586/2.5226 | 6.0578 | 21.2292 | 0.6952 | 18.1609 | 2.2274/2.4991 |
| AniPortrait (2024) | 3.0734 | 20.2589 | 0.6429 | 21.5914 | 3.2615/3.3014 | 2.6863 | 22.4807 | 0.6968 | 12.2362 | 2.2683/2.7352 |
| EDTalk (2024) | 8.0570 | 22.4354 | 0.7251 | 21.9435 | 2.8209/2.4370 | 6.3703 | 22.6067 | 0.7400 | 19.6080 | 2.2614/2.4689 |
| Echomimic (2025) | 5.3461 | 21.6390 | 0.6978 | 13.9435 | 2.4156/2.7941 | 4.5033 | 22.3503 | 0.6952 | 11.9544 | 2.2285/2.8633 |
| HalloV2 (2025) | 6.3832 | 21.4575 | 0.6779 | 15.6245 | 2.3489/2.5880 | 5.0140 | 23.2052 | 0.7129 | 10.7165 | 2.2149/2.5266 |
| Sonic (2025) | 8.0988 | 21.1874 | 0.7118 | 14.2623 | 2.5822/2.4025 | **6.8620** | 23.0787 | 0.7341 | 9.9440 | 1.9454/**2.3638** |
| DICE-Talk (2025) | 7.3073 | 19.7293 | 0.6279 | 27.9495 | 3.1125/3.3559 | 5.7601 | 21.4570 | 0.6675 | 17.8824 | 2.8910/3.2486 |
| HalloV1* (2024) | 4.9512 | 22.0258 | 0.7101 | 13.0673 | 2.5016/2.5885 | 4.5161 | 23.2809 | 0.7074 | 10.0336 | 2.1814/2.6313 |
| AUHead (HalloV1) | 6.0201 | 22.0132 | 0.7113 | 12.8421 | 2.3836/2.4595 | 4.7100 | 23.0818 | 0.7201 | 9.7086 | 2.2964/2.5337 |
| MEMO* (2025) | 6.9885 | 23.1910 | 0.7345 | 11.1237 | 2.0684/2.2473 | 6.0922 | 24.2808 | 0.7410 | 8.3881 | 1.9678/2.4296 |
| AUHead (MEMO) | 6.6311 | **23.3466** | **0.7395** | 10.9671 | **1.8608/2.1604** | 6.2050 | **24.2912** | 0.7413 | 8.2361 | **1.9313**/2.3991 |

**Qualitative Comparison** As demonstrated in Fig. 4, we conducted qualitative comparisons with state-of-the-art methods, including AniPortrait (Wei et al., 2024), Echomimic (Chen et al., 2024), HalloV1 (Xu et al., 2024), and MEMO (Zheng et al., 2024), on both the MEAD (Wang et al., 2020) and CREMA (Cao et al., 2014) datasets. While baseline models often suffer from blurred textures, distorted shapes, or emotionally flat expressions, AUHead produces clear, expressive, and visually coherent results. These empirical results show that integrating AU guidance into diffusion models is both effective and practical for expressive talking head synthesis.

To further demonstrate the effectiveness of AU-guided generation, we compare AUHead and the MEMO baseline under identical conditions: the same audio, identity image, and selected frames. The AUHead model uses AU features regressed from Qwen-Audio-Chat as guidance. As shown in Fig. 5, AUHead produces more vivid and nuanced facial expressions, even without explicit emotion instructions. The results reveal finer expression details, including accurate lip movements and vivid eye dynamics, demonstrating that AU conditioning can drive realistic facial animation. More detailed examples can be found in Supplementary Section B.

To evaluate whether the generated expressions align with AU guidance, we present video frames alongside their corresponding 2D AU representations, including 2D keypoint-based landmark (LMK) and 2D rendering of mesh (RoM). As shown in Fig. 6, the generated expressions closely match the input AU features, indicating accurate emotional representation. The results show that 2D AU representations offer effective emotion encoding and reliable guidance for facial animation.

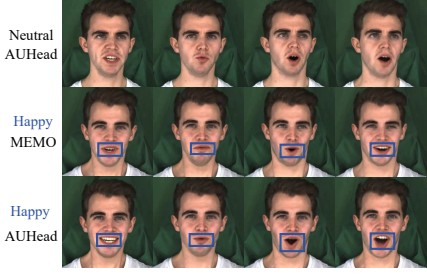

Figure 5: Facial animation comparison under neutral audio using AU from ALM.

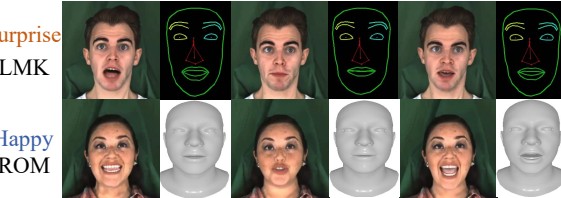

Figure 6: Visualization of generated frames and their corresponding AU-based 2D representations.

To further validate the generalization ability and temporal stability of AUHead, Fig. 7 presents several 10-second video examples generated on unseen data. Each row corresponds to a different audio

input, and each frame is randomly sampled from the corresponding second of the generated video. This setting allows us to examine both the consistency within long sequences and the model's robustness across diverse conditions. As shown in the figure, AUHead successfully handles different visual styles, including sketch, oil-painting, and realistic faces, while producing fine-grained and temporally coherent facial expressions. These results demonstrate that AUHead maintains high synthesis quality, identity preservation, and stable motion dynamics over extended time spans.

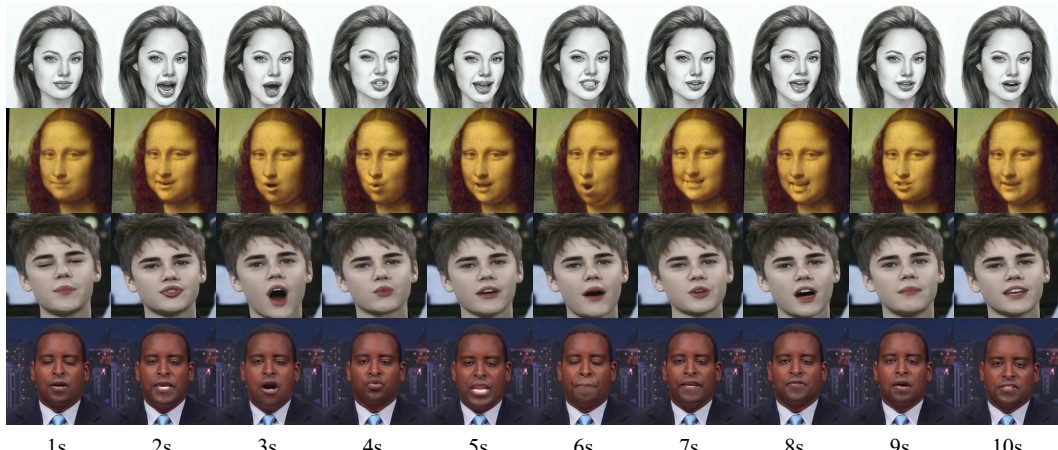

1s    2s    3s    4s    5s    6s    7s    8s    9s    10s

Figure 7: Visualization of AUHead's generalization across 10-second sequences. Each row corresponds to a unique combination of unseen audio and a new target identity. For each sequence, one frame is randomly sampled from each second of the generated video. The examples cover three visual styles, line-art sketches, oil-painting portraits, and realistic face images (Zhang et al., 2021) to illustrate the model's behavior under varied input domains.

### 4.3.1 USER STUDY

We conducted a user study to evaluate the perceptual quality of the generated talking-face videos. Evaluation samples consisted of 32 clips randomly selected from the MEAD test set, covering 4 identities and 8 emotions. Each participant compared results from AUHead and the state-of-the-art model HalloV2 (Cui et al., 2024a) under the same conditions. The

Table 4: User study evaluating the quality and emotional expressiveness of the generated talking heads. The better results are highlighted in **bold**.

| User Preference | HalloV2 | AUHead | Same |
|---|---|---|---|
| Emotional Expression | 18.88% | **64.63**% | 16.49% |
| Video Quality | 21.28% | **63.63**% | 15.09% |
| Audio-Lip Sync | 13.75% | **71.00**% | 15.25% |
| Overall Performance | 16.13% | **67.75**% | 16.12% |

study followed a blind setting, where participants were not informed of the method names. For each video pair, participants were asked to choose which result was better or indicate if there was no significant difference, based on four aspects: emotional expression, video quality, audio–lip synchronization, and overall performance. A total of 25 participants rated all video pairs, so that every pair received the same number of ratings. As shown in Table 4, AUHead consistently received the highest preference scores across all metrics, confirming its advantage in generating realistic, expressive, and well-synchronized talking-face videos. Participants also mentioned that AUHead delivered clear and coherent lip synchronization.

## 5 CONCLUSION

In this paper, we propose AUHead, a two-stage audio-driven talking head generation framework that uses AU representations as an intermediate bridge. In the first stage, fine-grained AU information is disentangled from audio signals using an ALM, enabling structured and interpretable control. In the second stage, the AU features drive the generation of realistic and emotionally expressive facial animations with consistent identity. Results on the MEAD and CREMA datasets show that AUHead achieves superior performance compared to existing methods, highlighting the effectiveness of AUs as a controllable and emotion-aware representation. Beyond performance, our work suggests that introducing interpretable intermediate spaces, like AUs, can offer a general strategy for more controllable and reliable cross-modal generation. In future work, we aim to extend AUHead to handle in-the-wild scenarios with diverse head poses and backgrounds.

## ACKNOWLEDGMENTS

This work was supported by the National Natural Science Foundation of China (62320106007, 62236006, 62521007) and the financial support of the State Key Laboratory of Communication Content Cognition. The first author would like to thank the China Scholarship Council (CSC) for the financial support during the doctoral research visit at the National University of Singapore.

## ETHICS STATEMENT

Our work on talking head generation makes it easier to create expressive and accessible visual content, with potential benefits for communication, education, and creative applications. At the same time, it poses risks such as possible misuse for deepfakes, biases in generated faces, and the environmental impact of large-scale training. We also recognize potential concerns regarding privacy and security when handling facial data. Addressing these challenges requires responsible dataset curation, fairness evaluation, clear documentation of research practices, and careful deployment.

## REPRODUCIBILITY STATEMENT

To promote reproducibility, we provide a repository link (https://github.com/laura990501/AUHead_ICLR) containing the implementation used in our experiments. In addition, both the main paper and the appendix detail the model architecture, training objectives, and evaluation settings. Together, these resources make it possible for others to replicate our work and verify the reported results.

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

## A   DETAILED INTRODUCTION TO FACIAL ACTION UNIT

Facial Action Units (AUs) are defined in the Facial Action Coding System (FACS), which was originally developed by Ekman & Friesen (1978). FACS provides a standardized framework for describing facial muscle movements. It decomposes facial expressions into 44 individual AUs, each corresponding to specific muscle activations. These units can appear independently or in combination to form complex expressions. To simplify expression modeling, the researchers redefined and selected 24 representative AUs commonly observed in facial expressions (Yan et al., 2019; Gan et al., 2022). A complete list of these 24 AUs is provided in Table 5. In FEAFA Yan et al. (2019); Gan et al. (2022), each AU is annotated with a continuous value ranging from 0 to 1, where 0 indicates no activation and 1 indicates maximum activation. As shown in Fig. 8, each AU is visually illustrated across different intensity levels, providing clear examples of activation changes across frames.

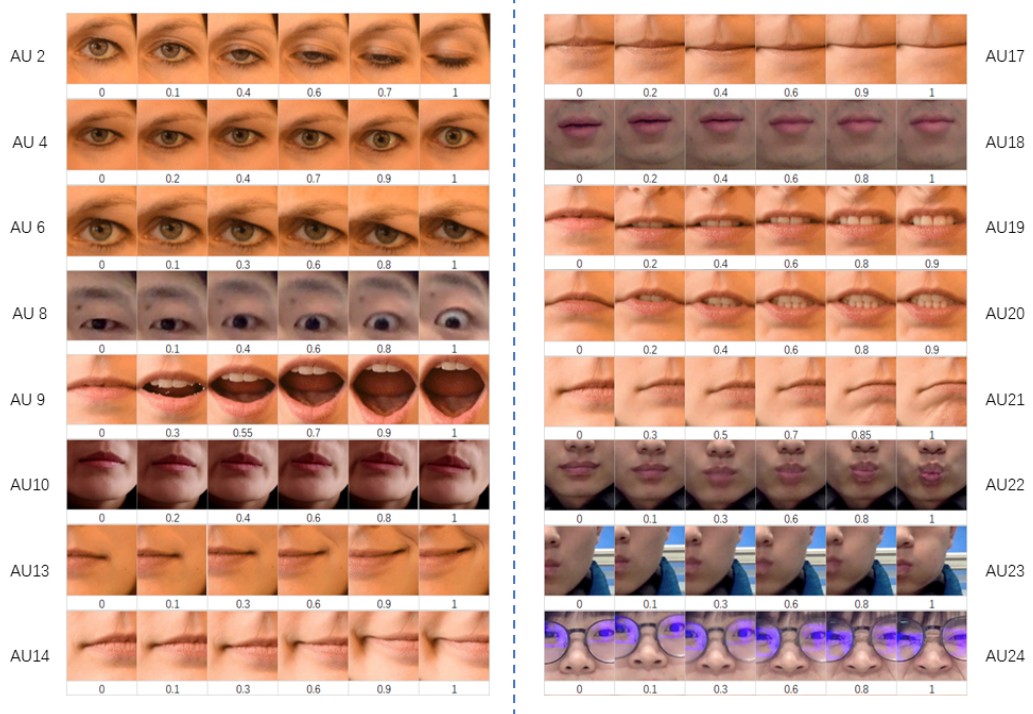

Figure 8: Visual examples of AU activations at different intensity levels in the FEAFA+ dataset. Each row corresponds to one AU, and columns show variations from neutral (*left*) to highly activated (*right*).

To ensure the quality of our AU data, we use an interactive AU validation tool during data construction. As shown in Fig. 9, this tool displays the aligned AU values, corresponding face images, and their 2D mesh projections for each frame. Although we do not manually annotate AUs, this visual interface allows us to carefully inspect whether the automatically extracted AU values align with the visual facial changes.

## B   IMPLEMENTATION DETAILS OF EMOTION GUIDED GENERATION

This section provides additional details and analysis corresponding to the emotion guided generation setting discussed in Section Experiment Fig. 5 and Fig. 10, where AUHead is compared to the MEMO baseline under identical input conditions. Specifically, both models receive the same neutral audio and a reference face image reflecting the target emotion. For AUHead, although the audio lacks emotional cues, we leverage the structure of the MEAD dataset, which includes repeated sentences performed with different emotional expressions. Using this property, we extract sparse AU

Table 5: Explanation of redefined AUs from the FEAFA+ dataset.

| FEAFA's Definition of AUs | |
| --- | --- |
| AU1: Left Eye Close | AU2: Right Eye Close |
| AU3: Left Upper Lid Raiser | AU4: Right Upper Lid Raiser |
| AU5: Left Brow Lowerer | AU6: Right Brow Lowerer |
| AU7: Left Outer Brow Raiser | AU8: Right Outer Brow Raiser |
| AU9: Jaw Drop | AU10: Left Jaw Sideways |
| AU11: Right Jaw Sideways | AU12: Left Lip Corner Puller |
| AU13: Right Lip Corner Puller | AU14: Left Lip Stretcher |
| AU15: Right Lip Stretcher | AU16: Upper Lip Suck |
| AU17: Lower Lip Suck | AU18: Jaw Thrust |
| AU19: Upper Lip Raiser | AU20: Lower Lip Depressor |
| AU21: Chin Raiser | AU22: Lip Pucker |
| AU23: Puff | AD24: Nose Wrinkler |

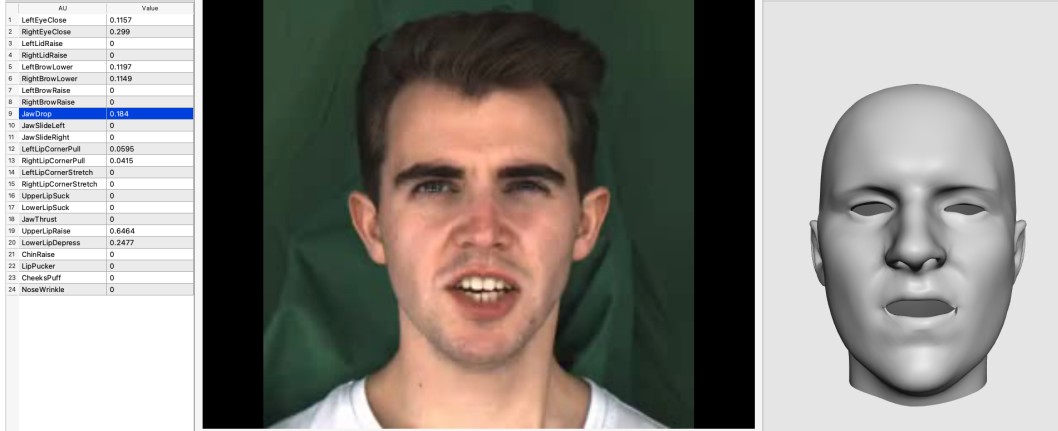

Figure 9: Frame level AU verification interface. Each frame displays the annotated AU values, the original face image, and its mesh based projection, enabling cross-modal consistency checking.

sequences from emotional versions of the same sentence via Qwen-Audio-Chat (Stage 1). These sequences are then temporally aligned with the neutral audio through linear interpolation and reshaped into 2D AU representations, which serve as AU guidance input to AUHead.

After generation, we extract the same set of keyframes from both models for visual comparison. It is important to note that the reference image plays a dominant role in talking head generation, as emphasized in the MEMO framework (Zheng et al., 2024). Nevertheless, AU representations offer complementary guidance that modulates subtle expression dynamics. They help the model generate more natural and emotionally aligned movements. Compared to MEMO, AUHead produces clearer lip shapes, more accurate brow motions, and realistic eye activity. Moreover, AUHead with emotional AU guidance shows clearer and more emotionally rich expressions than the neutral AU setting, further highlighting the benefit of AU features during generation.

## C PROMPT TEMPLATE

In Stage 1, we fine-tune the Qwen-Audio-Chat model using supervised instruction-following data. Each prompt consists of an audio file and a text instruction. The text part describes how the audio is processed (including the frame rate), lists the definition and order of all AUs, explains the format of AU values, and finally asks the model to classify the emotion and predict the AU sequence for each frame. Below is a real example prompt used in training:

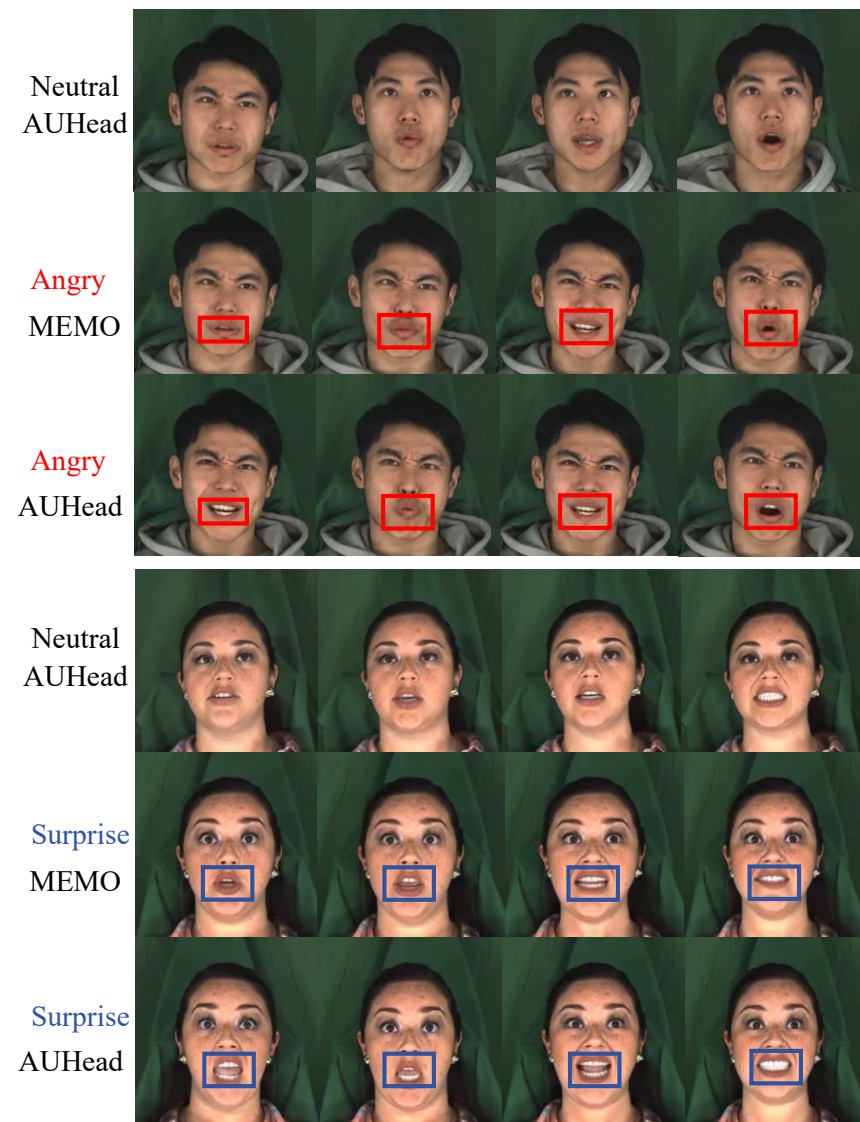

Figure 10: Facial animation comparison under neutral audio using AU from Qwen-Audio-Chat.

**Qwen-Audio-Chat Input Example**

```
{"messages": [
 {"role": "user", "audio": "data/MEAD_Audio_Train/M013_sur_3_014.wav", "content": "For each
     16kHz audio, split the waveform into frames of 3200 samples (5 fps). Each frame
     produces a 24-dimensional AU vector, with components AU0 to AU23 representing facial
     muscle activations in this fixed order: AU0 left eye closure; AU1 right eye closure;
     AU2 left lid raise; AU3 right lid raise; AU4 left brow lower; AU5 right brow lower;
     AU6 left brow raise; AU7 right brow raise; AU8 jaw-driven mouth opening; AU9 lower
     lip slide (left); AU10 lower lip slide (right); AU11 left lip corner raise; AU12
     right lip corner raise; AU13 left lip corner stretch; AU14 right lip corner stretch;
     AU15 upper lip suck; AU16 lower lip suck; AU17 jaw thrust; AU18 upper lip raise; AU19
     lower lip depress; AU20 chin raise; AU21 lip pucker; AU22 cheek puff; and AU23 nose
     wrinkle. Each AU value is between 0 and 1 and must be formatted to two decimal places
     (that is, write only the decimal point and two digits - for example, \".12\" for
     0.12). For each audio segment, record only the AUs that are activated along with
     their values; for example, [(0, .12), (1, .10)] means only AU0 is 0.12 and AU1 is
     0.10 while the others remain untriggered. What is the emotion of the current audio?
     What is the AU sequence of the current audio?"},
 {"role": "assistant", "content": "surprise, [[[2, 1.0], [3, 1.0], [6, 1.0], [7, 1.0], [8,
     0.52], [18, 0.17], [19, 0.11]], [[2, 1.0], [3, 1.0], [6, 1.0], [7, 1.0], [18, 0.19],
     [19, 0.08]], ... ]}
]}
```

This prompt enables the model to perform both emotion classification and AU sequence regression in a unified framework.

# D ADDITIONAL QUALITATIVE RESULTS

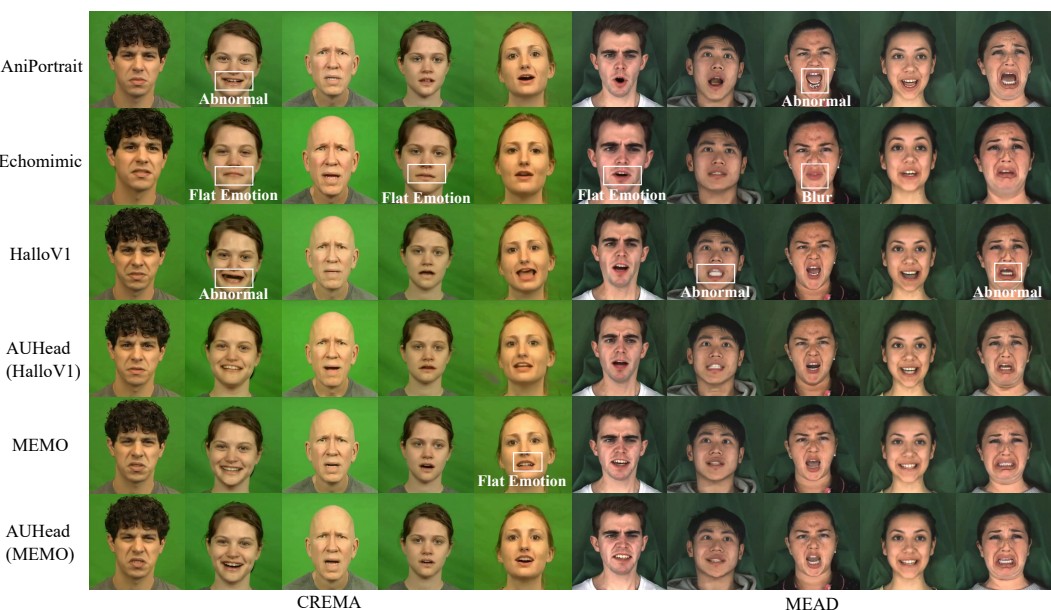

Figure 11: Qualitative comparison with SOTA methods on MEAD and CREMA.

Fig. 11 presents an extended visual comparison between AUHead and multiple SOTA methods on the MEAD and CREMA datasets. This figure complements the qualitative results shown in the main paper by including more models and a wider range of cases. We also annotate noticeable artifacts in the baseline outputs, such as blurry textures, unnatural expressions, or incorrect mouth shapes. In contrast, AUHead consistently produces clearer, more expressive, and visually coherent results, highlighting the benefits of AU-guided generation.

In addition to still-frame comparisons, we provide supplementary video demonstrations in the directory named AUHead_Supplementary_Material. It contains two folders. The first, compare_with_SOTA_method, includes video comparisons between AUHead and existing SOTA methods using identical audio and reference images. These clips offer a more intuitive view of AUHead's superior temporal consistency and expression quality. In addition to the video demonstrations, we further provide static frame comparisons extracted from these videos (see Figs. 14, 16, 13, 15, and 17). These visualizations serve as representative snapshots of the dynamic results, allowing for a clearer inspection of fine-grained details. Across diverse subjects and emotions, AU-Head produces frames with more natural and expressive facial dynamics, particularly in the subtle motion of the eyes and brows. Compared with baselines that often exhibit artifacts such as blurriness or rigid expressions, AUHead consistently delivers higher-quality outputs that preserve both emotional fidelity and identity stability. The second folder, Emotion, shows emotional generation results using neutral audio combined with AU features and reference images corresponding to different emotions. These examples demonstrate that AUHead is capable of generating expressive talking-head videos with strong emotional fidelity and stable identity preservation.

# E THE USE OF LARGE LANGUAGE MODELS

We declare that large language models (LLMs) were employed to assist with the refinement of this manuscript, specifically, for grammar checking, language polishing, and improving the clarity and fluency of the text. Additionally, LLMs were used in a limited capacity for minor debugging and syntactic correction of code snippets included in the work.

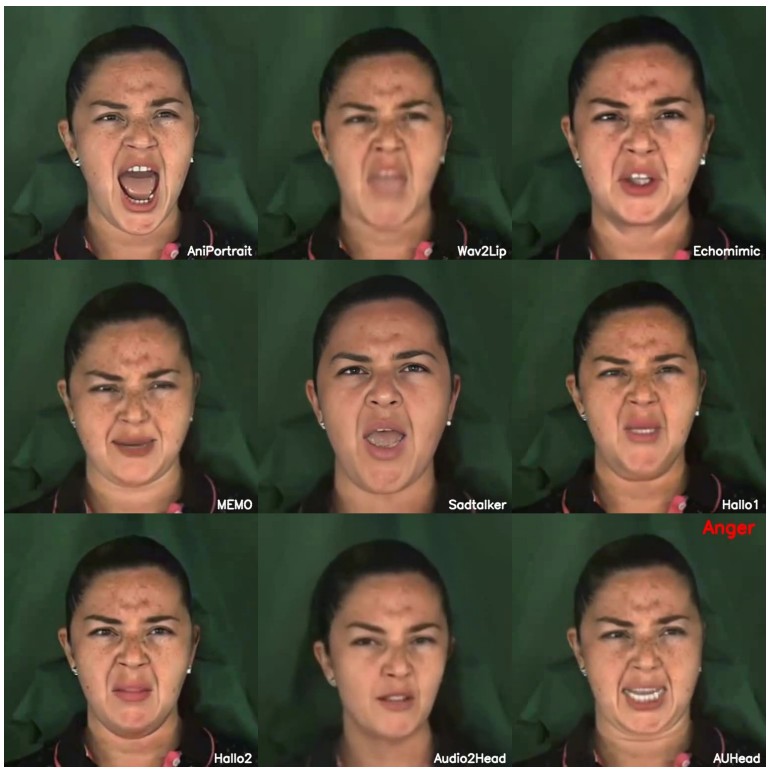

Figure 12: Qualitative comparison with SOTA methods on MEAD: results of subject W009 with *Angry* expression.

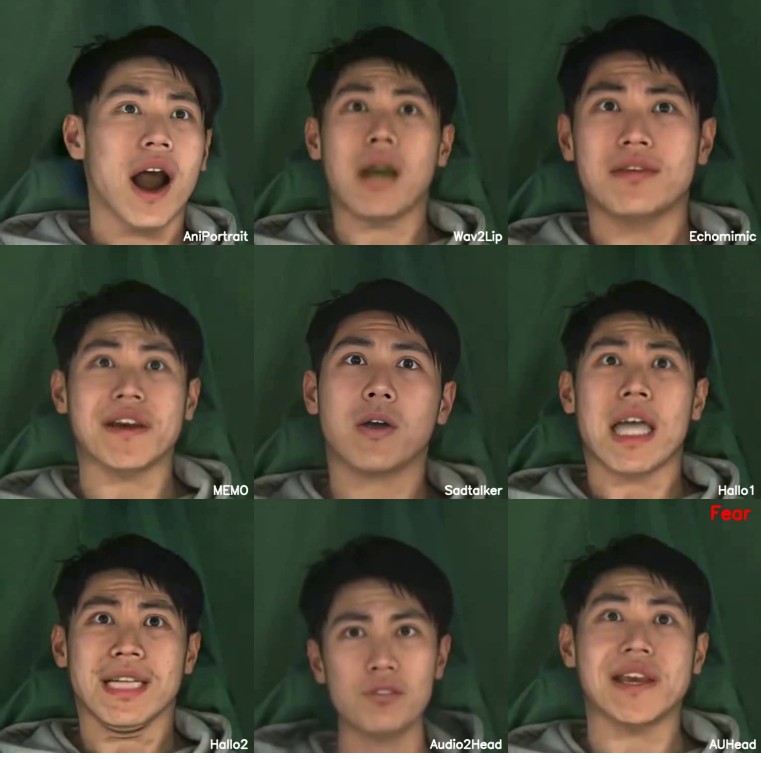

Figure 13: Qualitative comparison with SOTA methods on MEAD: results of subject M030 with *Fear* expression.

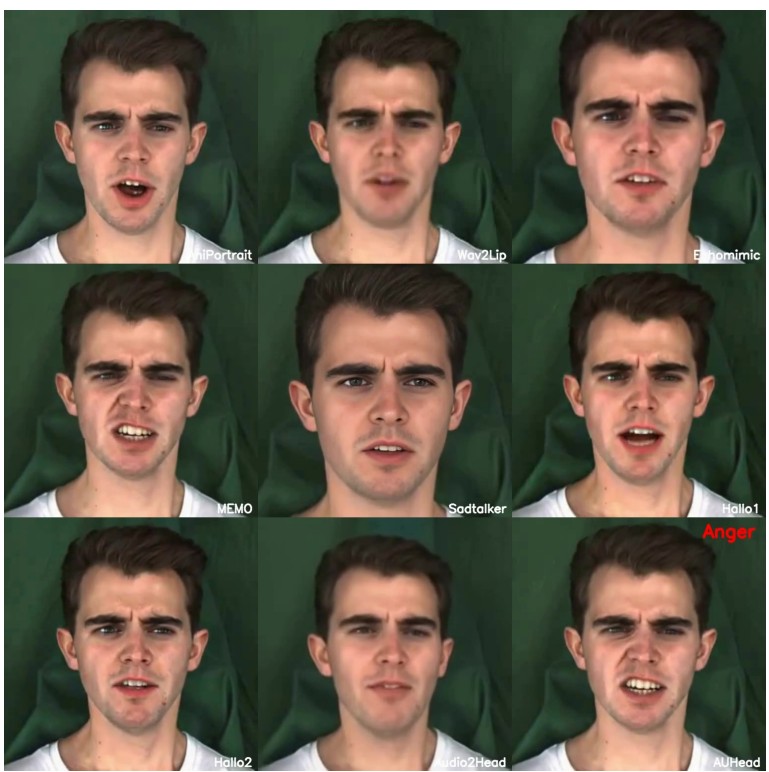

Figure 14: Qualitative comparison with SOTA methods on MEAD: results of subject M003 with *angry* expression.

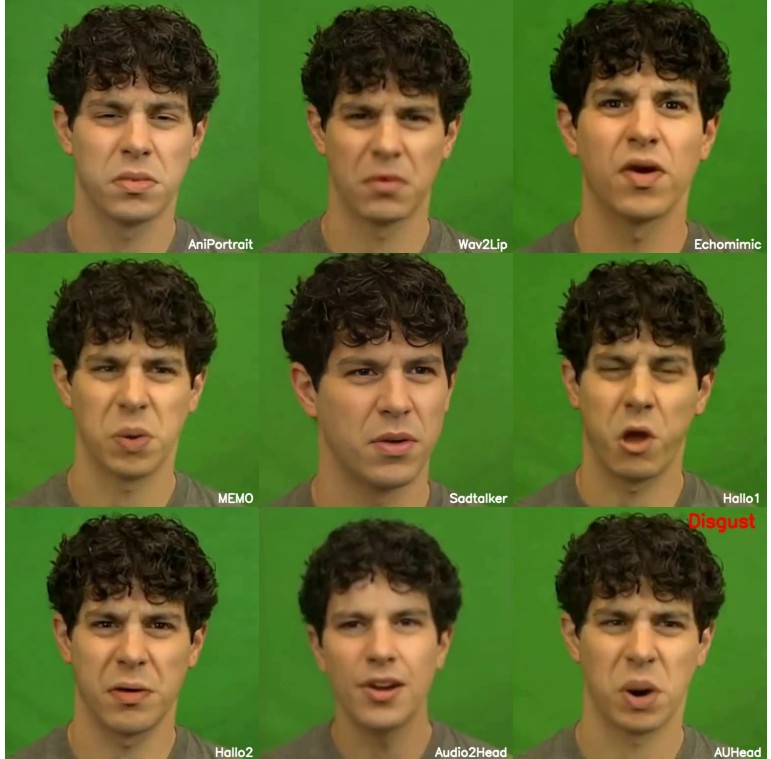

Figure 15: Qualitative comparison with SOTA methods on CREMA: results of subject 1033 with *Disgusted* expression.

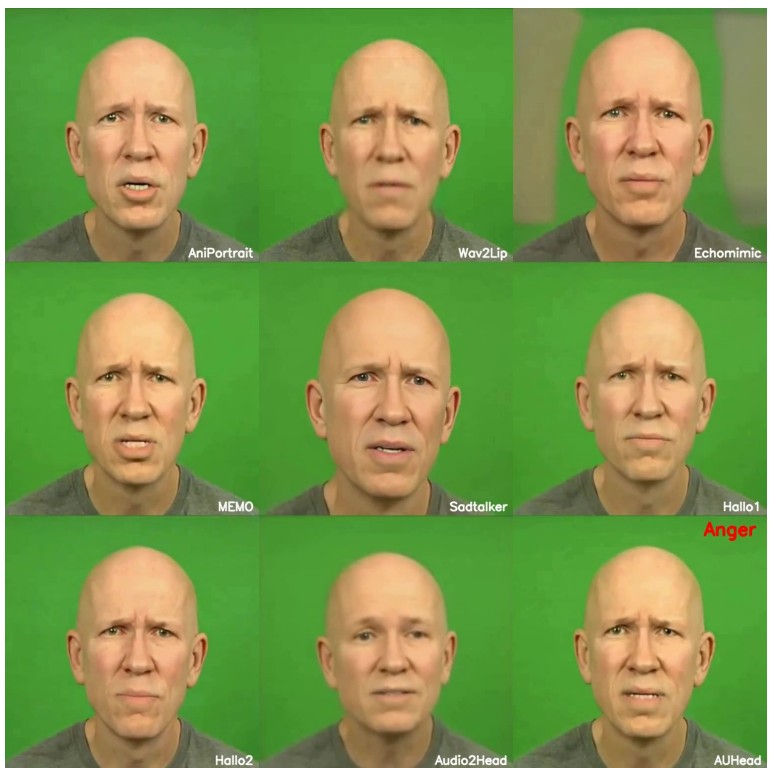

Figure 16: Qualitative comparison with SOTA methods on CREMA: results of subject 1062 with *Angry* expression.

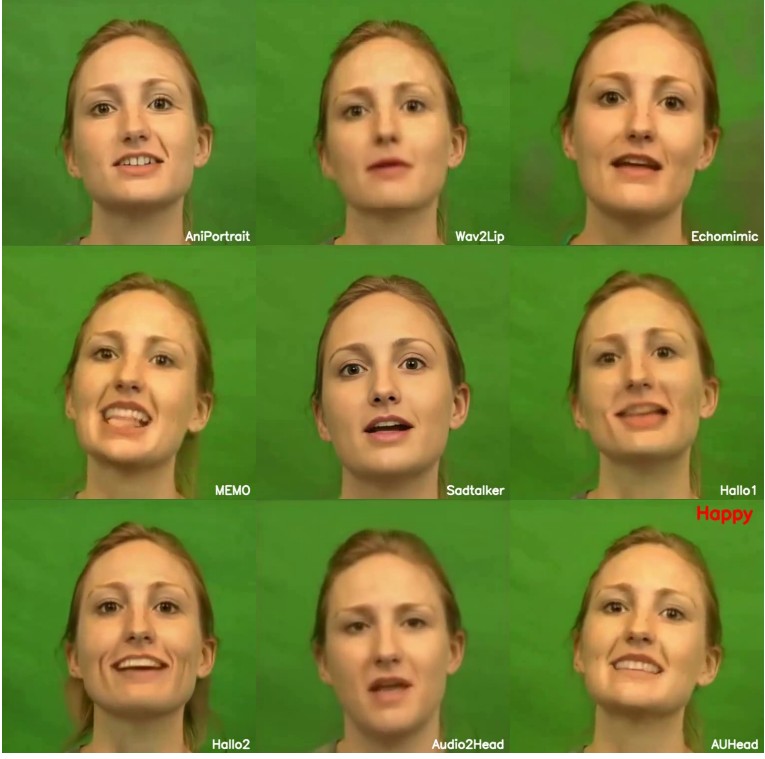

Figure 17: Qualitative comparison with SOTA methods on CREMA: results of subject 1089 with *Happy* expression.

