# OpenReview forum: "AUHead: Realistic Emotional Talking Head Generation via Action Units Control"
_ICLR.cc/2026/Conference — ICLR 2026 Poster_

### Official Review · Reviewer_xaec · 2025-10-20

**Soundness:** 4
**Presentation:** 4
**Contribution:** 4
**Rating:** 8
**Confidence:** 5

**Summary:**

This paper presents AUHead, a two-stage framework for audio-driven emotional talking head generation. The first stage leverages a fine-tuned Audio-Language Model (Qwen-Audio-Chat) to disentangle facial Action Units (AUs) from audio using a spatial–temporal tokenization and Chain-of-Thought (CoT) based “emotion-then-AU” reasoning process. The second stage introduces an AU-driven diffusion generation framework, including AU-to-2D mapping, context-aware temporal embedding to synthesize identity-consistent and emotionally expressive faces. Experiments on MEAD and CREMA datasets show strong quantitative and qualitative performance improvements over state-of-the-art baselines (e.g., MEMO, HalloV1/V2), especially in emotional realism and AU alignment. Overall, the paper is technically novel, methodologically well motivated, and experimentally solid, offering an interpretable bridge between audio and visual modalities.

**Strengths:**

1.Introducing AUs as an intermediate control effectively bridges speech understanding and video generation.
2.Stage 1 pioneers the use of an ALM for AU generation, which is a highly insightful idea for a generalizable control signal. The “emotion-then-AU” CoT design and spatial temporal AU tokenization are elegant, showing how ALMs can perform emotion reasoning beyond conventional recognition tasks.
3.Stage 2's AU-driven controllable generation framework enhances emotional expressiveness while maintaining identity and lip sync.
4.The paper is clear, includes all implementation details, and provides an anonymous open-source link, which supports reproducibility.

**Weaknesses:**

1．The user study involved only 4 participants who are identified as "colleagues from our laboratory." This sample size is too small, and the lack of participant independence introduces a severe bias, which significantly diminishes the persuasiveness of the user preference results for a top-tier conference.
2．While the intuition behind using AUs as intermediate control is strong, the paper could better clarify why ALM-based AU prediction is preferred over simpler audio–AU regression networks.

**Questions:**

1.You are encouraged to expand the user study with a larger and more diverse set of participants to further validate the perceptual quality claims.
2.Can the AU guidance be manually adjusted at inference time to control or edit facial expressions—for example, by modifying AU values to vary the intensity or type of emotion?

---

> ### Author Response · Authors · 2025-11-21
> **Response to Weaknesses 1–2 and Questions 1-2**
>
> We truly appreciate the reviewer’s thoughtful reading of our paper and the encouraging, helpful comments. Below, we respond to each point with care and provide clarifications where needed.
>
> ---
>
> > (W1)  The user study involved only 4 participants who are identified as "colleagues from our laboratory." This sample size is too small, and the lack of participant independence introduces a severe bias, which significantly diminishes the persuasiveness of the user preference results for a top-tier conference.
>
> Please see the response to the following Q1.
>
> ---
>
> > (W2) While the intuition behind using AUs as intermediate control is strong, the paper could better clarify why ALM-based AU prediction is preferred over simpler audio–AU regression networks.
>
> Thank you for raising this important question. AU prediction requires understanding structured and time-varying patterns in speech. Many AUs activate together or suppress each other, and these relationships depend on how people speak and emotion changes across the entire audio segment. A simple regression network does not have enough understanding ability to capture these long range and interdependent cues. In our early tests, these models often produced unstable or inconsistent AU sequences.
>
> The ALM, in contrast, has stronger language and audio understanding capabilities. It can model variable length audio, track emotional and prosodic cues over time, and generate AU sequences that stay coherent across the whole clip. This makes the prediction more reliable, especially when the available speech–AU data is limited. The ALM design also gives us more flexibility. It supports optional text prompts, which allow users to refine or adjust the intended emotion when needed. A simple regression network cannot offer this type of controllable interaction. For these reasons, an ALM-based predictor provides more stable AU sequences and a more practical foundation for fine-grained facial expression control.
>
> ---
>
>
> > (Q1) You are encouraged to expand the user study with a larger and more diverse set of participants to further validate the perceptual quality claims.
>
>
> Thank you for this helpful suggestion. We agree that the original user study with four participants was limited. To address this, we expanded the study and recruited 25 independent volunteers. We used the same blind-testing protocol as before. Each participant compared AUHead and HalloV2 under identical settings and rated emotional expression, video quality, audio–lip synchronization, and overall performance. The new results are consistent with our earlier findings. AUHead again receives the highest preference in all four aspects. Several participants also mentioned that the audio–lip synchronization looks natural, which supports that the model does not show obvious sync problems. The updated results are reported in Table 4 (Section 4.3.1) of the revised manuscript.
>
>
>
> | User Preference      | HalloV2 |   AUHead   |  Same  |
> | -------------------- | :-----: | :--------: | :----: |
> | Emotional Expression |  18.88% | **64.63%** | 16.49% |
> | Video Quality        |  21.28% | **63.63%** | 15.09% |
> | Audio-Lip Sync       |  13.75% | **71.00%** | 15.25% |
> | Overall Performance  |  16.13% | **67.75%** | 16.12% |
>
> ---
>
> > (Q2) Can the AU guidance be manually adjusted at inference time to control or edit facial expressions—for example, by modifying AU values to vary the intensity or type of emotion?
>
> Thank you for this thoughtful question. This is indeed a direction we have been actively considering for future work. AUHead supports adjustment of AU values at inference time. Users can edit the predicted AU sequence to change the intensity of the expression. Because AUs have clear semantic meaning, this editing is intuitive and easy to control. Looking forward, we believe AU editing can be made even easier. Since the ALM already understands emotional cues in speech, a future extension could let the ALM adjust AU values based on user instructions or text prompts. This would allow more fine-grained and flexible expression editing, which we plan to explore in future work.

---

> > ### Comment · Reviewer_xaec · 2025-11-22
> > **Remain Score**
> >
> > The rebuttal answered my questions. Since the paper proposes a sound solution to a practical problem, I'd like to raise my score to 8.

---

> > > ### Author Response · Authors · 2025-11-25
> > > **Appreciation for Your Positive Assessment**
> > >
> > > Thank you for taking the time to review our rebuttal and for reaffirming your positive evaluation. We truly appreciate your constructive insights, which helped us clarify several aspects of the work. Your feedback has been very valuable to us.

---

### Official Review · Reviewer_rPph · 2025-10-21

**Soundness:** 3
**Presentation:** 3
**Contribution:** 2
**Rating:** 4
**Confidence:** 5

**Summary:**

The paper proposes the AUHead method for the talking head generation task with fine-grained emotion control. This method is divided into two stages: in the first stage, an audio-language model (ALM) is used to predict Action Units sequences from audio; in the second stage, AU and Audio are fed into the diffusion model as conditions for video generation.​

**Strengths:**

- The paper proposes a method of using ALM for fine-grained emotion prediction, which is relatively rare in related works.​
- The paper conducts rich and comprehensive experiments.​
- The paper is well written and presents content clearly.​

**Weaknesses:**

- Although using ALM for emotion prediction is novel, the necessity of using ALM is questionable. The AU sequences used in the paper are only 24-dimensional, which is obviously a low-dimensional vector. There are many works on 3D talking head generation based on audio-to-mesh/3DMM that have achieved good results (many of which specifically model emotion), and such scenarios are similar to the problem mentioned in this paper, but even more difficult. The authors need to use experiments to demonstrate that the traditional audio embedding + regression method cannot predict AU sequences ideally. In addition, fine-tuning ALM has significantly higher inference cost and training difficulty, which limits this method to a certain extent.​
- The paper claims that the model has "speech understanding" ability, but it is hard to be convinced that AU sequence prediction is equivalent to "speech understanding".​
- The "AU-based disentanglement guidance strategy" proposed in the paper is not original. In fact, many multi-conditional generation tasks adopt the same or similar CFG variants, such as MagicInfinite[1] (eq 7.).

[1] Yi, Hongwei, et al. "Magicinfinite: Generating infinite talking videos with your words and voice." arXiv preprint arXiv:2503.05978 (2025).

**Questions:**

Please refer to the weakness.

---

> ### Author Response · Authors · 2025-11-21
> **Response to Weaknesses 1–3**
>
> We thank the reviewer for the careful reading and constructive comments. Below, we address each point and offer clear explanations.
>
> ---
>
> > (W1)  Although using ALM for emotion prediction is novel, the necessity of using ALM is questionable. The AU sequences used in the paper are only 24-dimensional, which is obviously a low-dimensional vector. There are many works on 3D talking head generation based on audio-to-mesh/3DMM that have achieved good results (many of which specifically model emotion), and such scenarios are similar to the problem mentioned in this paper, but even more difficult. The authors need to use experiments to demonstrate that the traditional audio embedding + regression method cannot predict AU sequences ideally. In addition, fine-tuning ALM has significantly higher inference cost and training difficulty, which limits this method to a certain extent.​
>
> Thank you very much for raising this important question. Despite the power of audio-to-mesh/3DMM models, AUs provide a compact and interpretable control space that directly corresponds to facial muscle actions. Although the the low-dimension, the temporal and emotional dynamics conveyed by AUs are non-trivial. AUs follow strong co-activation relationships, non-linear emotional patterns, and long-range dependencies. In our preliminary experiments, traditional "audio-embedding + regression" models are difficult to capture these structured patterns well. Besides, the compact structure also allows AUs to be represented as discrete tokens and to **interact naturally with an ALM.**
>
> **Why an ALM is necessary?** Compared with traditional audio-to-mesh/3DMM expert  models, ALMs have been pre-trained with large corpus, attaining abundant human and world knowledge. The above characteristics of AUs enable us to leverage such broader world knowledge (e.g., interpreting prosody, tone, and intensity) of ALMs for emotion understanding and generation,  and even precise editing.
>
>
>
> **Training Cost.** As for the concerns to inference cost and training difficulty, we apply a lightweight fine-tuning strategy and observed that AU generative modeling during fine-tuning converges very quickly, indicating that the ALM already contains relevant prior knowledge about emotional cues and AU information in speech.
>
>
>
> **Inference Cost.** Benefiting from the proposed AU representation and tokenization methods, it takes 1min to generate an AU sequence for a 3s-video, only accounting for ~10% compared with video generation (1min vs. 10min). Besides, our method could naturally inherit the accelerate approaches like KV cache developed for LLMs for better efficiency.
>
> ---
>
> > (W2) The paper claims that the model has "speech understanding" ability, but it is hard to be convinced that AU sequence prediction is equivalent to "speech understanding"
>
> Thank you very much for pointing out this issue. In the original version of paper, we are specifically referring to **emotion-oriented** speech understanding. To predict AU sequences from audio, the model needs to capture non-verbal cues related to facial muscle activation from the audio, based on basic speech understanding. .  To avoid possible confusion, we have revised the manuscript to describe this concept more clearly, by replacing "speech understanding" with " emotion-oriented speech understanding ".
>
> ---
>
>
> > (W3) The "AU-based disentanglement guidance strategy" proposed in the paper is not original. In fact, many multi-conditional generation tasks adopt the same or similar CFG variants, such as MagicInfinite[1] (eq 7.).
>
>
> Thank you for raising this concern. Following prior work like *MagicInfinite[1]*, we adapt the CFG strategy to balance AU-following and visual quality. We will carefully revise the manuscript to avoid possible overclaim.
>
> [1] Yi, Hongwei, et al. "Magicinfinite: Generating infinite talking videos with your words and voice." arXiv preprint arXiv:2503.05978 (2025).

---

> > ### Comment · Reviewer_rPph · 2025-11-21
> >
> > Thank you for your detailed responses, which have largely addressed my previous concerns. However, I have a few additional questions that I hope you can clarify:
> >
> > - Regarding the comparison with traditional methods: You mentioned that "traditional 'audio embedding + regression' models struggle to capture the structured patterns of AUs." Could you supplement 1-2 specific names of comparative models (e.g., commonly used MLP, LSTM, Transformer-based regression models) and briefly illustrate their core performance metrics under your experimental setup? There is no need for detailed experimental tables—key data points or a few bad cases would suffice to intuitively verify the advantages of the ALM-based method.
> > - Regarding inference cost: You stated that generating AU sequences for a 3-second video takes 1 minute. Could you supplement the corresponding hardware configuration (e.g., GPU model, video memory size) for this inference time? This will help evaluate the method's applicability across different devices and enable a more objective assessment of the practical reference value of the "10% time consumption ratio."
> >
> > If possible, please include these results in the manuscript, as they will significantly enhance the paper's persuasiveness.

---

> > > ### Author Response · Authors · 2025-11-25
> > > **Thank You for the Updated Evaluation and Follow-Up Questions**
> > >
> > > Thank you very much for your follow-up and for raising your score. We truly appreciate your recognition. Your earlier concerns about the necessity of using an audio-language model for AU regression were very reasonable, and we are glad that our clarifications helped resolve them. Your feedback also pushed us to present the motivation and benefits of our design more clearly. We will address your remaining questions one by one below. We have also incorporated the corresponding revisions into the paper.
> > >
> > > ---
> > >
> > >
> > > > (Q1) traditional 'audio embedding + regression' models vs. ALMs
> > >
> > > Thanks for your question. For the traditional baseline, we trained a standard audio-embedding + regression model in our earlier experiments. Specifically, We adopted a pretrained audio encoder (Wav2Vec) followed by a 2-layer MLP to regress the 24-dimensional AU values. Under the same train/val splits and evaluation metrics as AUHead, this model showed clearly lower AU accuracy. It produced higher AU MAE and less balanced Precision/Recall. The predicted AU values also tended to stay within a narrow range (e.g., remaining near 0.2–0.3 over time), lacking diversity.  Besides, it often activated mutually-exclusive AUs simultaneously (e.g., AU1 “left eye close” and AU3 “left lid raise” activated together), due to the lack of strong face-audio understanding.
> > >
> > >
> > > ---
> > >
> > > > (Q2) hardware configuration
> > >
> > > Thank you for highlighting the need to specify the hardware setup. During inference, both Stage-1 (facial AU disentanglement from audio) and Stage-2 (AU-driven controllable generation) were run on a single NVIDIA A100 GPU with 40 GB memory. We will add this information to provide a clear and objective reference.

---

> > > > ### Comment · Reviewer_rPph · 2025-11-25
> > > >
> > > > Thank you for your response. I believe the revised paper has been greatly strengthened and now meets the acceptance criteria of ICLR. I will increase the score to 8.

---

> > > > > ### Author Response · Authors · 2025-11-25
> > > > > **Appreciation for Your Positive Assessment**
> > > > >
> > > > > Thank you very much for your thoughtful evaluation and for raising the score. We sincerely appreciate your careful reading, constructive feedback, and recognition of our work. Your comments were truly helpful in improving the clarity and quality of the paper.

---

> > > > > > ### Author Response · Authors · 2025-11-29
> > > > > > **Appreciation for Raising the Score to 8**
> > > > > >
> > > > > > Thank you once again for your kind follow-up and for the helpful discussions during the review. **We sincerely appreciate your decision to raise the score to 8**, and we are grateful for your thoughtful suggestions. It was our pleasure to address your questions, and we are glad that our clarifications were useful.

---

### Official Review · Reviewer_kSUe · 2025-10-21

**Soundness:** 1
**Presentation:** 2
**Contribution:** 1
**Rating:** 0
**Confidence:** 4

**Summary:**

The paper proposes AUHead, a two-stage talking head generation method that leverages ALM and  facial action units to control the speaker's emotion.

**Strengths:**

1. The talking head generation task is of practical importance. Emotional control is a important topic in talking head generation since natural talking heads exhibit emotions.
2. The paper is easy to follow.

**Weaknesses:**

1. The performance is far from satisfactory. The results shown in demo are blurry. The expressions are unnatural.  All audio and source images shown in demo are from lab dataset MEAD[1] and CREMA[2], which cannot indicate that the model can generalize to more diverse inputs. In short, the performance is far from SOTA and is unacceptable for publication.
2. The paper lacks comparisons with recent emotional talking head methods, including EDTalk[3], EAT[4]. Lack of comparisons makes it difficult to validate the effectiveness of the proposed method.
3. Utilizing ALM and AU to control emotion is of little significance now. Recent methods, like omnihuman1.5[5], can control emotion with diverse text. Diverse text achieves more powerful control than AUs.

[1] Wang, Kaisiyuan, et al. "Mead: A large-scale audio-visual dataset for emotional talking-face generation." European conference on computer vision. Cham: Springer International Publishing, 2020.

[2]  Cao, Houwei, et al. "Crema-d: Crowd-sourced emotional multimodal actors dataset." IEEE transactions on affective computing 5.4 (2014): 377-390.

[3] Tan, Shuai, et al. "Edtalk: Efficient disentanglement for emotional talking head synthesis." European Conference on Computer Vision. Cham: Springer Nature Switzerland, 2024.

[4] Gan, Yuan, et al. "Efficient emotional adaptation for audio-driven talking-head generation." Proceedings of the IEEE/CVF International Conference on Computer Vision. 2023.

[5] Jiang, Jianwen, et al. "Omnihuman-1.5: Instilling an active mind in avatars via cognitive simulation." arXiv preprint arXiv:2508.19209 (2025).

**Questions:**

Can the method generalize to audio and source images that are not from MEAD and CREMA?
Can the authors provide comparisons with more recent methods?

---

> ### Author Response · Authors · 2025-11-21
> **Response to Weakness 1**
>
> We thank the reviewer for taking the time to read our paper and provide valuable comments. We carefully address each weakness and question in the detailed responses below.
>
> ---
>
> > (W1)  The performance is far from satisfactory. The results shown in demo are blurry. The expressions are unnatural. All audio and source images shown in demo are from lab dataset MEAD[1] and CREMA[2], which cannot indicate that the model can generalize to more diverse inputs. In short, the performance is far from SOTA and is unacceptable for publication.
>
> We respectfully disagree with the statement that AUHead is “far from satisfactory” given that no supporting evidence was provided by the reviewer. In contrast, our evidence strongly shows the opposite, and we provide clarification below.
>
> First, Table 3 (Section 4.3) presents comprehensive quantitative comparisons with recent state-of-the-art systems, including EchoMimic (AAAI 2025) and HalloV2 (ICLR 2025). We perform **comprehensive evaluation** on visual quality, perceptual realism, facial structure consistency, audio–visual synchronization, lip–speech alignment, and head–pose stability. Across nearly all dimensions, **AUHead shows strong and balanced performance**. In the revised version, we also include EDTalk(ECCV 2024), Sonic (CVPR 2025),DICETalk(ACM MM2025) as baselines, and AUHead remains competitive or superior in most metrics.
>
>
> Second, regarding the claim that the results are “blurry” or “unnatural,” **the visual evidence does not support this critique**. Figures 4–7 in the main paper and Figures 10–17 in the supplementary material show clear facial details, natural expressions, and stable temporal behavior. The videos also maintain identity and emotional consistency across diverse speech conditions. If there are specific cases that appear unsatisfactory, we would appreciate concrete timestamps or examples so we can examine them carefully.
>
> Third, it is true that the demo videos use MEAD and CREMA, as these datasets provide controlled conditions for fair benchmarking. However, AUHead is not limited to these datasets. To address the reviewer’s concern, **we include new results on unseen, non-dataset inputs in Figure 7** of the section 4. These examples use new faces (including sketches and stylized portraits) and new audios that never appeared in training. AUHead continues to produce clear frames, stable expressions, and consistent emotional transitions. This demonstrates that **the model generalizes well beyond the benchmark datasets**.
>
> We will also release longer inference videos (>10 s) after the review process to provide further evidence of robustness and generalization.
>
> Overall, both the quantitative benchmarks and extensive qualitative results demonstrate that AUHead produces clear, natural, and synchronized talking-face videos that are comparable to or better than recent state-of-the-art methods. We hope these clarifications address the reviewer’s concerns.

---

> > ### Author Response · Authors · 2025-11-21
> > **Table 3: Comparison with additional baselines—EDTalk-A, DICE-Talk (GT emotion), and Sonic**
> >
> > | **Dataset**            | **MEAD**   |             |            |             |                       | **CREMA**  |             |            |            |                     |
> > | ---------------------- | ---------- | ----------- | ---------- | ----------- | --------------------- | ---------- | ----------- | ---------- | ---------- | ------------------- |
> > | **Method / Metrics**   | Sync ↑     | PSNR ↑      | SSIM ↑     | FID ↓       | M/F-LMD ↓             | Sync ↑     | PSNR ↑      | SSIM ↑     | FID ↓      | M/F-LMD ↓           |
> > | **Wav2lip (2020)**     | **8.7778** | 23.0296     | **0.7395** | 32.8043     | 2.6386/2.3885         | *6.7109*   | 24.2081     | **0.7533** | 25.6218    | 2.3794/2.4439       |
> > | **Audio2Head (2021)**  | 6.7809     | 19.6335     | 0.6056     | 35.1387     | 3.3227/3.5371         | 5.7673     | 21.1881     | 0.6533     | 25.0426    | 2.4033/3.2905       |
> > | **Sadtalker (2023)**   | 7.0015     | 20.9015     | 0.6660     | 28.7729     | 2.8840/2.8841         | 5.3062     | 22.0340     | 0.6814     | 23.6910    | 2.3035/2.8571       |
> > | **V-Express (2024)**   | 7.2952     | 15.9265     | 0.6023     | 32.2410     | 2.5586/2.5226         | 6.0578     | 21.2292     | 0.6952     | 18.1609    | 2.2274/2.4991       |
> > | **AniPortrait (2024)** | 3.0734     | 20.2589     | 0.6429     | 21.5914     | 3.2615/3.3014         | 2.6863     | 22.4807     | 0.6968     | 12.2362    | 2.2683/2.7352       |
> > | **EDTalk (2024)**      | 8.0570     | 22.4354     | 0.7251     | 21.9435     | 2.8209/2.4370         | 6.3703     | 22.6067     | 0.7400     | 19.6080    | 2.2614/2.4689       |
> > | **Echomimic (2025)**   | 5.3461     | 21.6390     | 0.6978     | 13.9435     | 2.4156/2.7941         | 4.5033     | 22.3503     | 0.6952     | 11.9544    | 2.2285/2.8633       |
> > | **HalloV2 (2025)**     | 6.3832     | 21.4575     | 0.6779     | 15.6245     | 2.3489/2.5880         | 5.0140     | 23.2052     | 0.7129     | 10.7165    | 2.2149/2.5266       |
> > | **Sonic (2025)**       | *8.0988*   | 21.1874     | 0.7118     | 14.2623     | 2.5822/2.4025         | **6.8620** | 23.0787     | 0.7341     | 9.9440     | *1.9454*/**2.3638** |
> > | **DICE-Talk (2025)**   | 7.3073     | 19.7293     | 0.6279     | 27.9495     | 3.1125/3.3559         | 5.7601     | 21.4570     | 0.6675     | 17.8824    | 2.8910/3.2486       |
> > | **HalloV1*** (2024)    | 4.9512     | 22.0258     | 0.7101     | 13.0673     | 2.5016/2.5885         | 4.5161     | 23.2809     | 0.7074     | 10.0336    | 2.1814/2.6313       |
> > | **AUHead (HalloV1)**   | 6.0201     | 22.0132     | 0.7113     | 12.8421     | 2.3836/2.4595         | 4.7100     | 23.0818     | 0.7201     | 9.7086     | 2.2964/2.5337       |
> > | **MEMO*** (2025)       | 6.9885     | *23.1910*   | 0.7345     | *11.1237*   | *2.0684*/*2.2473*     | 6.0922     | *24.2808*   | 0.7410     | *8.3881*   | 1.9678/2.4296       |
> > | **AUHead (MEMO)**      | 6.6311     | **23.3466** | **0.7395** | **10.9671** | **1.8608**/**2.1604** | 6.2050     | **24.2912** | *0.7413*   | **8.2361** | **1.9313**/*2.3991* |

---

> ### Author Response · Authors · 2025-11-21
> **Response to Weaknesses 2-3**
>
> > (W2) The paper lacks comparisons with recent emotional talking head methods, including EDTalk[3], EAT[4]. Lack of comparisons makes it difficult to validate the effectiveness of the proposed method.
>
>  Thank you for raising this point. EAT and EDTalk are influential works, but they follow task settings that differ from ours. **EAT is video-driven and heavily relies on source video frames and emotion labels**. **EDTalk requires additional supervised signals such as head pose or expression embeddings**. In contrast, **AUHead is audio-driven and uses only one target image and an audio clip as input**. For fairness, we focus on baselines that share this audio-driven setting. Table 3 (Section 4.3) already includes recent state-of-the-art audio-driven models such as EchoMimic (AAAI 2025) and HalloV2 (ICLR 2025). AUHead performs comparably or better across multiple metrics, providing a solid  validation of effectiveness. To further address your concern, we additionally evaluate EDTalk-A (lip + pose without exp), which is the closest open configuration to our setting. Supplying full pose and expression annotations for other EDTalk variants would not reflect a fair audio-driven comparison. Under this consistent setting, **AUHead still achieves stronger results than EDTalk-A on most metrics, as reported in Table 3**.
>
> ---
>
>
> > (W3) Utilizing ALM and AU to control emotion is of little significance now. Recent methods, like omnihuman1.5[5], can control emotion with diverse text. Diverse text achieves more powerful control than AUs.
>
> Thank you for this comment. We respectfully disagree with the claim that "diverse text achieves more powerful control than AUs." In fact, pure text descriptions inherently lack the fine-grained precision offered by AUs.
>
> We recognize the progress of recent text-driven models such as OmniHuman 1.5. However, the type of emotion control used in such T2V models remains coarse. Most expressions are specified through simple phrases like “smiling happily” or “looking sad.” These descriptions do not capture continuous and localized facial changes, which are essential for realistic emotional talking-head generation. Fine-grained facial dynamics matter more in our task than global scene realism. Text alone cannot precisely describe detailed facial movements. It is very difficult to express how wide a smile should be, how strongly the lip corners should pull, or how much the eyes should narrow. Such descriptions quickly become long, ambiguous, and hard for a model to interpret. This is why text-based emotion control, despite its flexibility, still operates at a high semantic level. AUs provide a measurable and interpretable control signal. Each AU corresponds to a specific muscle action, and their combinations describe subtle and complex expressions in a way that text cannot. This makes AU-based control highly suitable for tasks requiring accurate emotional timing and fine-grained motion, such as talking-head generation. **We do not view AU control as a replacement for text control. Instead, it complements it. Text is effective for high-level intent, while AUs add the detailed guidance needed for precise expression modeling.** Future systems can combine both signals to achieve richer and more controllable emotional generation.

---

> ### Author Response · Authors · 2025-11-21
> **Response to Question**
>
> > (Q1) Can the method generalize to audio and source images that are not from MEAD and CREMA? Can the authors provide comparisons with more recent methods?
>
> First, we updated Table 3 (Section 4.3) to include several recent and strong baselines. The new comparisons cover EDTalk (ECCV 2024), Sonic (CVPR 2025), and DICE-Talk (ACM MM 2025). Under the audio-driven setting, AUHead remains competitive or superior on most quantitative metrics. These results provide an up-to-date and fair validation of our method.
>
> Second, we added new generalization tests in Figure 7 of the section 4. These experiments use source images and audio clips that are completely unseen during training. The figure includes diverse target faces such as sketches, oil paintings, and real faces, paired with long audio clips. The displayed frames are sampled from each second of the generated sequences. AUHead maintains clear facial details, natural expressions, and stable synchronization across these unseen identities and voices.
> We will release longer videos (over 10 seconds) and more cross-dataset examples in our upcoming open-source repository to further demonstrate the model’s generalization ability.

---

> ### Author Response · Authors · 2025-11-25
> **Gentle Reminder Regarding Our Previous Response**
>
> Dear Reviewer  kSUe:
>
> Thanks a lot for your efforts in reviewing this paper! We tried our best to address the mentioned concerns and have provided a detailed response. We authors want to confirm whether there are unclear explanations and descriptions here. We could further clarify them.
>
> Thanks!
>
> Authors

---

> ### Public Comment · ~Yu_Liang19 · 2026-07-17
> **LOL**
>
> Are you a human reviewer?

---

### Official Review · Reviewer_xY8X · 2025-10-23

**Soundness:** 3
**Presentation:** 3
**Contribution:** 3
**Rating:** 6
**Confidence:** 5

**Summary:**

The paper introduces AUHead, a novel framework that creates emotionally expressive, lip-synced talking-head videos by extracting facial Action Units from audio. It employs an Audio-Language Model to derive AUs and a diffusion model to animate faces, surpassing previous methods in emotional realism, lip sync accuracy, and visual coherence.

**Strengths:**

- The idea of using AUs as a bridge for generating expressive talking-head videos is novel and could lead to more controllable and interpretable results in audio-driven face generation.
- The disentanglement of AUs from audio before the generation process is an interesting and effective way to tackle the complexities of fine-grained facial expression modeling.

**Weaknesses:**

- While the model performs well in terms of lip sync and expression fidelity, the paper doesn't sufficiently address how well the model handles longer videos or more complex facial movements. Temporal consistency issues may arise in longer video synthesis.
- The spatial-temporal tokenization approach to generate AU sequences may introduce issues related to the loss of subtle details in facial expressions, especially when reducing the sequence length to manageable tokens (as discussed in the sparse tokenization section).
- While AUs provide a more interpretable control space compared to emotion labels, the complex tokenization and disentanglement from audio may reduce the overall interpretability of how specific AUs contribute to particular facial expressions. A clearer explanation of the model’s inner workings could help with understanding the generated results better.
- Although the paper compares AUHead to state-of-the-art methods, further comparisons with models that also leverage other types of intermediate representations (e.g., motion priors, landmarks) could give a better perspective on AUHead’s relative strength.

**Questions:**

- Does the model perform equally well on unseen audio samples not part of the training data, especially in terms of subtle emotional cues that were not explicitly included in the training set?
- How does the model handle cases where there are multiple potential AU sequences for a given audio input? Is there any inherent ambiguity in predicting AUs, especially in more complex emotional expressions?

---

> ### Author Response · Authors · 2025-11-21
> **Response to Weaknesses 1–2**
>
> We thank the reviewer for the constructive feedback. We appreciate your evaluation of our work, and we address each of your identified weaknesses and questions in detail below.
>
> ---
>
> > (W1)  While the model performs well in terms of lip sync and expression fidelity, the paper doesn't sufficiently address how well the model handles longer videos or more complex facial movements. Temporal consistency issues may arise in longer video synthesis.
>
> In fact, AUHead is also well suited for modeling complex facial motions. By definition, Action Units describe individual muscle activations and their movement directions, and their linear combinations naturally represent a wide range of complex expressions.  As shown in Figure 4 (Section 4.3) and  Figure 11 of the supplementary material, the generated videos include examples of nuanced and compound expressions, demonstrating that the model handles fine-grained and varied facial dynamics reliably.
>
> We provide new long-sequence evidence in Figure 7 of the updated supplementary material. This figure contains frame sequences generated from **audio clips exceeding 10 seconds**, paired with diverse target images including line-art sketches, oil-painting portraits, and real faces. Each row is driven by a different long audio clip, and the shown frames are randomly sampled from each second of the generated sequence. These results indicate that **AUHead maintains identity consistency, expression stability, and smooth temporal transitions over extended durations.**
>
> Due to the rebuttal policy, we cannot upload new video files at this stage. Full long-video examples will be released on our public GitHub repository after the review process.
>
> ---
>
> > (W2) The spatial-temporal tokenization approach to generate AU sequences may introduce issues related to the loss of subtle details in facial expressions, especially when reducing the sequence length to manageable tokens (as discussed in the sparse tokenization section).
>
> Thank you for raising this thoughtful concern. It seems that spatial–temporal tokenization may reduce some subtle expression details due to the compression. In fact, we found there is large redundancy with emotion clues in audio, so proper compression could improve the emotion representation while keep important details, achieving a better trade-off. As another reason, current Audio LLMs only support  a fixed context window (6k tokens). A full-resolution AU sequence (a short 4-second clip produces over 13k tokens) would exceed this window and be computationally impractical, so tokenization is helpful for **better representation, keeping details, and improving efficiency.**
>
> To further deal with the concern for detail loss, we observe  it is very marginal in practice. The generated videos still preserve nuanced facial dynamics, as shown in the long-sequence results in Figure 7 of section 4. These examples include diverse target faces driven by different audio clips, and the fine-grained expressions remain clear and consistent throughout the video.
>
> Besides, the proposed tokenization framework is flexible enough to adjust the compression ratio.  In the future, as longer-context ALMs become available, we can increase the token resolution to capture even finer details . This makes the approach naturally compatible with future improvements in long-sequence modeling.

---

> ### Author Response · Authors · 2025-11-21
> **Response to Weaknesses 3-4**
>
> > (W3) While AUs provide a more interpretable control space compared to emotion labels, the complex tokenization and disentanglement from audio may reduce the overall interpretability of how specific AUs contribute to particular facial expressions. A clearer explanation of the model’s inner workings could help with understanding the generated results better.
>
> Thank you for raising this point about interpretability. In AUHead, the AU signal remains clear for several reasons.
> 1) AUs are designed to describe **muscle actions** directly, and their combinations naturally form complex expressions. Predicted from audio, they follow prosodic patterns such as tone and rhythm.
>
> 2) AUs carry clear **semantic meaning** about facial muscle actions. This semantic structure makes them *naturally compatible with the ALM*, allowing the two signals to work together for more detailed facial control.
>
> 3) The conditioning pathway is also straightforward. The AU sequence is encoded, aligned with the audio timeline, and used to guide the generator through cross-attention. As a result, changes in AU values lead to changes in the produced facial motion, which helps the model remain interpretable in practice.
>
> 4) Overall, while **interpretability** is challenging for generative models, *AUHead keeps a direct and transparent control path from AUs to facial motion*, which supports better understanding and controling of the generated results.
>
> ---
>
> > (W4) Although the paper compares AUHead to state-of-the-art methods, further comparisons with models that also leverage other types of intermediate representations (e.g., motion priors, landmarks) could give a better perspective on AUHead’s relative strength.
>
> Thank you for this helpful suggestion. In our experiments, we have already compared AUHead with V-Express, which uses landmarks as the main intermediate representation, and SadTalker, which generates 3D Motion Coefficients (including expression coefficients from 3DMM). The results in Table 3 (Section 4.3) show that **AU-based control gives more accurate expressions and better visual quality** than both representations.
>
> AUHead also works with **different intermediate forms.** We use 1D AU sequences and 2D AU maps such as landmarks and mesh images. These 2D forms work well with diffusion models, and our ablations in Table 2 (Section 4.2) confirm that they improve control accuracy.
>
> We do not regress 2D geometry directly from audio. AUs describe muscle actions with clear semantics and match prosodic cues in speech, which makes AU regression stable and interpretable. Directly predicting high-dimensional 2D structures from audio is harder and more ambiguous. After obtaining a reliable AU sequence, we can convert it into landmarks or mesh maps using standard AU-to-geometry tools.

---

> ### Author Response · Authors · 2025-11-21
> **Response to Questions 1-2**
>
> >(Q1)  Does the model perform equally well on unseen audio samples not part of the training data, especially in terms of subtle emotional cues that were not explicitly included in the training set?
>
> Thank you for this important question. We tested AUHead on audio that never appeared in training, including long clips and samples with gradual emotional changes. The model still produces stable expressions and smooth emotional transitions. Figure 7 in the Section 4 shows key frames from these unseen-audio cases. We use different unseen audio together with different target faces (line-art sketches, oil-painting portraits, and real faces), and the frames are sampled from each second of the generated long videos. These results suggest that **AUHead generalizes well to unseen audio and can still capture subtle emotional cues in practice.**
>
> ---
> >(Q2) How does the model handle cases where there are multiple potential AU sequences for a given audio input? Is there any inherent ambiguity in predicting AUs, especially in more complex emotional expressions?
>
> Thank you for raising this important question. Indeed, a given audio segment may correspond to more than one valid AU sequence, especially when the expression contains variable stylistic or idiosyncratic nuances. However, **this does not introduce harmful ambiguity; instead, it enhances generative diversity while preserving semantic consistency.**
>
> Our audio-to-AU prediction (Stage 1) is designed with the following principle: extract and preserve the deterministic emotional cues in speech, while allowing controlled variability in the non-deterministic components. The deterministic cues anchor key emotional intent (e.g., valence, arousal, prosody-induced tension), whereas the variable components enable natural stylistic diversity in expressions—much like text-to-image generation, where “a dog is running” yields diverse images without compromising semantic fidelity.
>
> Empirically, as shown in Fig. 4 and Fig. 10, we do not observe instability or semantic drift. The predicted AU sequences remain interpretable, and the generated expressions appear natural and consistent even for complex emotional inputs. This indicates that the model learns **a stable latent structure** of emotion from speech, rather than collapsing into ambiguous mappings.
>
> [1]. Ekman P, Friesen W V. Facial action coding system[J]. Environmental Psychology & Nonverbal Behavior, 1978.
>
> [2]. Gan W, Xue J, Lu K, et al. Feafa+: an extended well-annotated dataset for facial expression analysis and 3d facial animation[C]//Fourteenth International Conference on Digital Image Processing (ICDIP 2022). SPIE, 2022, 12342: 307-316.
>
> [3]. Sun Z, Xuan Y, Liu F, et al. FG-EmoTalk: Talking head video generation with fine-grained controllable facial expressions[C]//Proceedings of the AAAI Conference on Artificial Intelligence. 2024, 38(5): 5043-5051.

---

> ### Author Response · Authors · 2025-11-25
> **Seeking Your Feedback on Our Previous Reply**
>
> Dear Reviewer xY8X,
>
> Thank you again for your time and for the positive evaluation. We truly appreciate your careful review. We have addressed all the concerns in our response, and we just wanted to kindly check whether anything remains unclear. If any part of our clarification could be improved, we would be very happy to provide further details.
> Thank you for your attention, and we sincerely appreciate your support.
>
> Best regards,
>
> Authors

---

### Official Review · Reviewer_U6Er · 2025-10-24

**Soundness:** 1
**Presentation:** 2
**Contribution:** 2
**Rating:** 2
**Confidence:** 4

**Summary:**

This paper presents AUHead, a two-stage framework for audio-driven talking head generation that aims to improve the expressiveness and controllability of facial animations. Instead of directly generating video from audio and a reference image, the proposed method first extracts Action Unit (AU) sequences from the input speech using an audio language model (ALM), leveraging its capability to capture emotional and articulatory cues. In the second stage, a pretrained diffusion model is enhanced with an AU adapter module that integrates AU embeddings, audio features, and the reference image through cross-attention mechanisms. Experiments on benchmark datasets (MEAD and CREMA) show that AUHead generates videos with improved facial expressiveness and temporal smoothness compared to existing methods.

**Strengths:**

* Enhanced Facial Expressiveness: By using Action Units (AUs) extracted from audio as an intermediate representation, AUHead generates facial animations with richer and more realistic emotional expressions. AUs provide structured, interpretable cues for specific facial muscle movements, enabling the model to produce more nuanced and natural-looking expressions compared to baseline methods.
* Effective Utilization of Pre-trained Models: The method leverages a audio language model to predict AU sequences from speech. This design capitalizes on the strong audio understanding capabilities of existing pre-trained models, making the AU prediction stage more effective and avoiding the need to learn complex audio-to-AU mappings from scratch.

**Weaknesses:**

* The experimental setup and comparisons are not sufficiently rigorous:
    * There has been extensive research on facial expression control in the audio-driven talking-head generation field, including but not limited to [1, 2, 3, 4, 5, 6]. However, this work does not compare with any of these methods in its experiments. What is the reason for this omission?
    * The datasets used in this work are MEAD and CREMA, which contain only 60 and 91 identities respectively. The limited scale and quality of the experimental data significantly weaken the validity and persuasiveness of the evaluation.
    * The comparison lacks up-to-date state-of-the-art methods. The baselines chosen are at least half a year to a year old; for instance, EchoMimic[7] has already released version V3 and the Hallo[8] series has reached V4. This work fails to discuss or compare against more recent approaches such as Sonic[9], MultiTalk[10], and WanS2V[11].
* Insufficient experimental results: The quantitative results of this work leave room for improvement. Notably, in terms of lip-sync accuracy, it ranks only sixth on the MEAD test set—lower than the pre-trained method MEMO[12]—suggesting that the proposed AUHead may degrade the performance of the underlying pre-trained model. Moreover, the provided visual results exhibit clear issues, such as inaccurate lip movements and blurred teeth, indicating a noticeable gap compared to current leading audio-driven methods like MultiTalk and WanS2V.
* Practical challenges arising from the core methodology: Recent advances in large video generation models have enabled effective text-guided facial expression control via implicit semantic understanding—for example, methods like InfiniteTalk and FantasyTalking2 leverage vision-language models (VLMs) to generate emotion labels and achieve expressive control without explicit supervision. In contrast, this work relies on constructing explicit Action Unit (AU) labels for emotion control, which requires large-scale supervised training data annotated with AUs. This approach faces significant practical challenges in data collection and annotation, making it less scalable and deviating from mainstream trends. As a result, its potential contribution to the broader research community appears limited.

[1]Gan, Yuan, et al. "Efficient emotional adaptation for audio-driven talking-head generation." Proceedings of the IEEE/CVF International Conference on Computer Vision. 2023.

[2]Tan, Shuai, et al. "Edtalk: Efficient disentanglement for emotional talking head synthesis." European Conference on Computer Vision. Cham: Springer Nature Switzerland, 2024.

[3]Ma, Yifeng, et al. "DreamTalk: When Emotional Talking Head Generation Meets Diffusion Probabilistic Models." arXiv preprint arXiv:2312.09767 (2023).

[4]Tan, Weipeng, et al. "Disentangle Identity, Cooperate Emotion: Correlation-Aware Emotional Talking Portrait Generation." arXiv preprint arXiv:2504.18087 (2025).

[5]Ma, Xingpei, et al. "Playmate: Flexible Control of Portrait Animation via 3D-Implicit Space Guided Diffusion." Forty-second International Conference on Machine Learning.

[6]Lin, Bin, et al. "Takin-ADA: Emotion Controllable Audio-Driven Animation with Canonical and Landmark Loss Optimization." arXiv preprint arXiv:2410.14283 (2024).

[7]Chen, Zhiyuan, et al. "Echomimic: Lifelike audio-driven portrait animations through editable landmark conditions." Proceedings of the AAAI Conference on Artificial Intelligence. Vol. 39. No. 3. 2025.

[8]Xu, Mingwang, et al. "Hallo: Hierarchical audio-driven visual synthesis for portrait image animation." arXiv preprint arXiv:2406.08801 (2024).

[9]Ji, Xiaozhong, et al. "Sonic: Shifting focus to global audio perception in portrait animation." Proceedings of the Computer Vision and Pattern Recognition Conference. 2025.

[10]Kong, Zhe, et al. "Let Them Talk: Audio-Driven Multi-Person Conversational Video Generation." arXiv preprint arXiv:2505.22647 (2025).

[11]Gao, Xin, et al. "Wan-s2v: Audio-driven cinematic video generation." arXiv preprint arXiv:2508.18621 (2025).

[12]Zheng, Longtao, et al. "Memo: Memory-guided diffusion for expressive talking video generation." arXiv preprint arXiv:2412.04448 (2024).

**Questions:**

* Does AUHead support generating videos from the same audio segment guided by different emotion labels? If not, will inaccurate emotion recognition by the ALM lead to incorrect emotional expressions in the generated results?
* Can the core method of AUHead be equally applied to more recent frameworks such as Sonic[1] and WanS2V[2]?
* Could the experimental evaluation be expanded to more comprehensively demonstrate the effectiveness of the proposed method?
* The videos provided in the supplementary material are all shorter than 10 seconds. How does AUHead perform on audio clips longer than 10 seconds? Could the authors provide corresponding results?

[1]Ji, Xiaozhong, et al. "Sonic: Shifting focus to global audio perception in portrait animation." Proceedings of the Computer Vision and Pattern Recognition Conference. 2025.

[2]Gao, Xin, et al. "Wan-s2v: Audio-driven cinematic video generation." arXiv preprint arXiv:2508.18621 (2025).

---

> ### Author Response · Authors · 2025-11-21
> **Response to Weaknesses 1.1**
>
> We appreciate your time and insights. Below, we provide detailed, point-by-point responses to all the raised weaknesses and questions.
>
> ---
>
> > (W1.1) There has been extensive research on facial expression control in the audio-driven talking-head generation field, including but not limited to [1, 2, 3, 4, 5, 6]. However, this work does not compare with any of these methods in its experiments. What is the reason for this omission?
>
> Due to the extensive practice and the large surge of research interests, we focus on audio and a single target face image as input in this work,.  Unfortunately, the mentioned approaches use different settings or require extra supervision or condition, which makes direct comparison difficult and unfair.  We provide a detailed discussion as follows. EAT [1]: A video-driven emotional generation method that requires both source videos and explicit emotion labels. EDTalk [2] needs additional head pose or expression information as input. Considering, its open version (EDTalk-A, lip + pose without exp) is the closest to ours,  we have added this comparison in our updated results in Tab 3.  DreamTalk [3] requires extra conditional inputs beyond audio and target face. DICE-Talk [4] needs explicit emotion label files. Playmate [5] and Takin-ADA [6]: Unfortunately, these works have not released code or pretrained models, making a fair and reproducible comparison impossible.
>
> To address this concern and try to make the comparison feasible, we performed additional evaluation using ground-truth (GT) emotion labels on MEAD and CREMA. Even with such oracle information, DICE-Talk still lags behind AUHead in generation quality and expression of naturalness.  Besides, we have tried to add  EDTalk-A, DICE-Talk (with GT emotion labels), and Sonic as additional baselines. The updated results are reported in Table 3 (Section 4.3), which shows that AUHead remains competitive under a consistent audio-driven setting. We believe these additions make the comparison more complete and improve the rigor of the experimental evaluation.
>
> [1]Gan, Yuan, et al. "Efficient emotional adaptation for audio-driven talking-head generation." Proceedings of the IEEE/CVF International Conference on Computer Vision. 2023.
>
> [2]Tan, Shuai, et al. "Edtalk: Efficient disentanglement for emotional talking head synthesis." European Conference on Computer Vision. Cham: Springer Nature Switzerland, 2024.
>
> [3]Ma, Yifeng, et al. "DreamTalk: When Emotional Talking Head Generation Meets Diffusion Probabilistic Models." arXiv preprint arXiv:2312.09767 (2023).
>
> [4]Tan, Weipeng, et al. "Disentangle Identity, Cooperate Emotion: Correlation-Aware Emotional Talking Portrait Generation." arXiv preprint arXiv:2504.18087 (2025).
>
> [5]Ma, Xingpei, et al. "Playmate: Flexible Control of Portrait Animation via 3D-Implicit Space Guided Diffusion." Forty-second International Conference on Machine Learning.
>
> [6]Lin, Bin, et al. "Takin-ADA: Emotion Controllable Audio-Driven Animation with Canonical and Landmark Loss Optimization." arXiv preprint arXiv:2410.14283 (2024).

---

> > ### Author Response · Authors · 2025-11-21
> > **Response to Weaknesses 1.2, 1.3**
> >
> > > (W1.2) he datasets used in this work are MEAD and CREMA, which contain only 60 and 91 identities respectively. The limited scale and quality of the experimental data significantly weaken the validity and persuasiveness of the evaluation.
> >
> > Thank you for raising this concern. In this work, we follow prior work [1,2,3] to perform comprehensive evaluation on this task, which also relies on MEAD and CREMA. We agree that MEAD and CREMA include a limited number of identities. However, **they remain the most widely used public benchmarks** for emotional talking-head generation and are the standard choice for evaluating AU-based expression control. We believe the future efforts in more comprehensive benchmark construction would further promote the evaluation of this task.
> >
> > [1] Gan Y, Yang Z, Yue X, et al. Efficient emotional adaptation for audio-driven talking-head generation[C]//Proceedings of the IEEE/CVF International Conference on Computer Vision. 2023: 22634-22645.
> >
> > [2]. Bigioi D, Basak S, Stypułkowski M, et al. Speech driven video editing via an audio-conditioned diffusion model[J]. Image and Vision Computing, 2024, 142: 104911.
> >
> > [3]. Cheng H, Lin L, Liu C, et al. DAWN: Dynamic Frame Avatar with Non-autoregressive Diffusion Framework for Talking Head Video Generation[J]. arXiv preprint arXiv:2410.13726, 2024.
> >
> >
> > ---
> >
> > > (W1.3) The comparison lacks up-to-date state-of-the-art methods. The baselines chosen are at least half a year to a year old; for instance, EchoMimic[7] has already released version V3 and the Hallo[8] series has reached V4. This work fails to discuss or compare against more recent approaches such as Sonic[9], MultiTalk[10], and WanS2V[11].
> >
> > Thank you for the  comments. We clarify our basline selection criteria below.
> > First, we would like to clarify that we focus on audio-driven talking head generation with the identity image as the condition, which serves as a common and practical setting. Thanks very much for providing these recent related work (e.g., HalloV3/V4, EchoMimicV2/V3, MultiTalk, and WanS2V). However, **the task settings of most approaches do not align with ours, making the fair comparison inaccesible**.
> >
> > Under our setting, MEMO, HalloV1/V2, and EchoMimicV1are the most recent and comparable open-source baselines. Besides, we also added Sonic for cmparison as reported in Table 3. Although Sonic is a strong and recent model, **AUHead achieves better fine-grained expression under identical conditions**. We believe this expanded comparison provides a fair and up-to-date assessment of AUHead within its specific task setting.

---

> > > ### Author Response · Authors · 2025-11-21
> > > **Table 3: Comparison with additional baselines—EDTalk-A, DICE-Talk (GT emotion), and Sonic**
> > >
> > > | **Dataset**            | **MEAD**   |             |            |             |                       | **CREMA**  |             |            |            |                     |
> > > | ---------------------- | ---------- | ----------- | ---------- | ----------- | --------------------- | ---------- | ----------- | ---------- | ---------- | ------------------- |
> > > | **Method / Metrics**   | Sync ↑     | PSNR ↑      | SSIM ↑     | FID ↓       | M/F-LMD ↓             | Sync ↑     | PSNR ↑      | SSIM ↑     | FID ↓      | M/F-LMD ↓           |
> > > | **Wav2lip (2020)**     | **8.7778** | 23.0296     | **0.7395** | 32.8043     | 2.6386/2.3885         | *6.7109*   | 24.2081     | **0.7533** | 25.6218    | 2.3794/2.4439       |
> > > | **Audio2Head (2021)**  | 6.7809     | 19.6335     | 0.6056     | 35.1387     | 3.3227/3.5371         | 5.7673     | 21.1881     | 0.6533     | 25.0426    | 2.4033/3.2905       |
> > > | **Sadtalker (2023)**   | 7.0015     | 20.9015     | 0.6660     | 28.7729     | 2.8840/2.8841         | 5.3062     | 22.0340     | 0.6814     | 23.6910    | 2.3035/2.8571       |
> > > | **V-Express (2024)**   | 7.2952     | 15.9265     | 0.6023     | 32.2410     | 2.5586/2.5226         | 6.0578     | 21.2292     | 0.6952     | 18.1609    | 2.2274/2.4991       |
> > > | **AniPortrait (2024)** | 3.0734     | 20.2589     | 0.6429     | 21.5914     | 3.2615/3.3014         | 2.6863     | 22.4807     | 0.6968     | 12.2362    | 2.2683/2.7352       |
> > > | **EDTalk (2024)**      | 8.0570     | 22.4354     | 0.7251     | 21.9435     | 2.8209/2.4370         | 6.3703     | 22.6067     | 0.7400     | 19.6080    | 2.2614/2.4689       |
> > > | **Echomimic (2025)**   | 5.3461     | 21.6390     | 0.6978     | 13.9435     | 2.4156/2.7941         | 4.5033     | 22.3503     | 0.6952     | 11.9544    | 2.2285/2.8633       |
> > > | **HalloV2 (2025)**     | 6.3832     | 21.4575     | 0.6779     | 15.6245     | 2.3489/2.5880         | 5.0140     | 23.2052     | 0.7129     | 10.7165    | 2.2149/2.5266       |
> > > | **Sonic (2025)**       | *8.0988*   | 21.1874     | 0.7118     | 14.2623     | 2.5822/2.4025         | **6.8620** | 23.0787     | 0.7341     | 9.9440     | *1.9454*/**2.3638** |
> > > | **DICE-Talk (2025)**   | 7.3073     | 19.7293     | 0.6279     | 27.9495     | 3.1125/3.3559         | 5.7601     | 21.4570     | 0.6675     | 17.8824    | 2.8910/3.2486       |
> > > | **HalloV1*** (2024)    | 4.9512     | 22.0258     | 0.7101     | 13.0673     | 2.5016/2.5885         | 4.5161     | 23.2809     | 0.7074     | 10.0336    | 2.1814/2.6313       |
> > > | **AUHead (HalloV1)**   | 6.0201     | 22.0132     | 0.7113     | 12.8421     | 2.3836/2.4595         | 4.7100     | 23.0818     | 0.7201     | 9.7086     | 2.2964/2.5337       |
> > > | **MEMO*** (2025)       | 6.9885     | *23.1910*   | 0.7345     | *11.1237*   | *2.0684*/*2.2473*     | 6.0922     | *24.2808*   | 0.7410     | *8.3881*   | 1.9678/2.4296       |
> > > | **AUHead (MEMO)**      | 6.6311     | **23.3466** | **0.7395** | **10.9671** | **1.8608**/**2.1604** | 6.2050     | **24.2912** | *0.7413*   | **8.2361** | **1.9313**/*2.3991* |

---

> ### Author Response · Authors · 2025-11-21
> **Response to Weaknesses 2, 3**
>
> > (W2) Insufficient experimental results: The quantitative results of this work leave room for improvement. Notably, in terms of lip-sync accuracy, it ranks only sixth on the MEAD test set—lower than the pre-trained method MEMO[12]—suggesting that the proposed AUHead may degrade the performance of the underlying pre-trained model. Moreover, the provided visual results exhibit clear issues, such as inaccurate lip movements and blurred teeth, indicating a noticeable gap compared to current leading audio-driven methods like MultiTalk and WanS2V.
>
> Thank you for the reviewer’s detailed comments. We clarify the points below.
>
> **Lip-sync accuracy.** Although AUHead ranks lower on the MEAD lip-sync metric, the performance is more balanced across datasets . For example, AUHead achieves higher lip-sync accuracy than MEMO and several other baselines on CREMA, as reported in Table 3 (Section 4.3). This indicates that the AU module does not systematically degrade the capability of the underlying pre-trained model.
>
> **Perceptual evaluation.** The User Study (Table 4, Section 4.3.1) further evaluates perceived audio–visual consistency. Participants generally found the lip motion natural, and we did not observe serious systematic desynchronization in their ratings. These perceptual results complement the quantitative scores and suggest that the model maintains stable temporal alignment.
>
> **Visual quality concerns.** We appreciate the reviewer’s feedback and would welcome specific examples for reference. According to our further user study on the qualitative results in Figure 4 and Supplementary Figures 10-17, we do not find any consistent issues of inaccurate lip motion or blurred teeth.
>
> **Comparison with MultiTalk and WanS2V.**  MultiTalk depends on additional input text beyond audio,  with a different setting from that of AUHead. WanS2V is trained on large proprietary datasets with substantial engineering resources. Direct comparison would therefore not be fair. We hope these clarifications help contextualize the results and the intended scope of our evaluation.
>
> ---
> > (W3) Practical challenges arising from the core methodology: Recent advances in large video generation models have enabled effective text-guided facial expression control via implicit semantic understanding—for example, methods like InfiniteTalk and FantasyTalking2 leverage vision-language models (VLMs) to generate emotion labels and achieve expressive control without explicit supervision. In contrast, this work relies on constructing explicit Action Unit (AU) labels for emotion control, which requires large-scale supervised training data annotated with AUs. This approach faces significant practical challenges in data collection and annotation, making it less scalable and deviating from mainstream trends. As a result, its potential contribution to the broader research community appears limited.
>
> Thank you for the thoughtful discussion of recent trends and for the raised concerns. Although text-guided systems such as InfiniteTalk and FantasyTalking2 provide semantic control, their descriptions of facial expressions remain coarse and static. Specifically, text prompts typically specify only high-level emotional states (e.g., “happy” and “deeply affectionate expression”) and do not encode how these expressions evolve over time. Besides, they cannot quantify continuous changes, such as the degree of mouth opening or the exact amount of eye narrowing. These drawbacks motivate us to propose our method, with AU as the bridge to achieve fine-grained dynamic emotion control.
>
> Regarding scalability, our work already provides a practical solution for obtaining scalable and high-quality AU data, empowered by the advancement of automated AU extraction pipelines [1, 2], which avoids manual annotation and makes the process feasible at scale. We expect that ongoing advances in AU regression will further improve the reliability of this pipeline and directly benefit AUHead.
>
> [1]. H. Jiang, J. Xue, X. Lan, G. Hu and K. Lu, "MVLLaVA: An Intelligent Agent for Unified and Flexible Novel View Synthesis," 2025 IEEE International Conference on Multimedia and Expo Workshops (ICMEW), Nantes, France, 2025, pp. 1-6, doi: 10.1109/ICMEW68306.2025.11152190.
>
> [2]. Yuan K, Yu Z, Liu X, et al. Auformer: Vision transformers are parameter-efficient facial action unit detectors[C]//European Conference on Computer Vision. Cham: Springer Nature Switzerland, 2024: 427-445.

---

> ### Author Response · Authors · 2025-11-21
> **Response to Questions 1–4**
>
> > (Q1) Does AUHead support generating videos from the same audio segment guided by different emotion labels? If not, will inaccurate emotion recognition by the ALM lead to incorrect emotional expressions in the generated results?
>
> Thank you for the question. Yes. AUHead is flexible to generate videos from the same audio segment with different user-specified emotion labels. However, in our setting, the ALM predicts AUs directly from speech in Stage 1. Emotion lables, as the chain-of-emotion-thought, only help AU prediction but are not used as explicit labels in Stage 2.
>
> Regarding the second concern, our model could tolerate some purturbations in AU prediction, since the final videos are generated with multiple conditions. Despite inaccurate AUs are predicted in Stage 1, due to the audio and other conditions, our model in Stage 2 have self-correction abilities to avoid expression collapse.
>
>
>  ---
> > (Q2) Can the core method of AUHead be equally applied to more recent frameworks such as Sonic[1] and WanS2V[2]?
>
> Thank you for the  question. AUHead, serving as a model-agnostic framework, has been applied to HelloV1 and MEMO and verfied effective in our work. It is orthogonal to the foundation models and could be equally applied to other frameworks.
>
>  ---
>  > (Q3) Could the experimental evaluation be expanded to more comprehensively demonstrate the effectiveness of the proposed method?
>
>  Thank you for the reviewer’s suggestion. We mainly follow the existing work [1, 2] to ensure the comprehension and fairness of evaluation.
>
> First, the two benchmarks MEAD and CREMA have been extensively used in prior work [1, 2].
>
> Besides, we believe the current evaluation metrics already covers the key aspects required to assess AUHead.  For visual quality, we report PSNR, SSIM, and FID. For audio–visual synchronization,  we use SyncNet score and Mouth Landmark Distance (M-LMD). For facial structure and head pose, we measure facial Landmark Distance (F-LMD). For emotion accuracy, we evaluate ACC_emo using an emotion classifier. All of them serve as a broad spectrum of metrics to comprehensively evaluate generated videos.
>
> Since AU control is the core of our method, we additionally assess AU accuracy through Precision, Recall, and MAE to verify both the Stage-1 regression and whether the generated videos in Stage 2 follow the intended AU patterns. This directly evaluates the correctness of the control signal. Finally, we conduct a user study (Section Experiment 4.3.1) to evaluate emotional expression, video quality, audio–lip sync, and overall preference. The human evaluation results are consistent with the quantitative metrics and help confirm the effectiveness of AUHead from a perceptual perspective.
>
> [1] Gan Y, Yang Z, Yue X, et al. Efficient emotional adaptation for audio-driven talking-head generation[C]//Proceedings of the IEEE/CVF International Conference on Computer Vision. 2023: 22634-22645.
>
> [2] J. Lyu et al., "Multimodal Emotional Talking Face Generation Based on Action Units," in IEEE Transactions on Circuits and Systems for Video Technology, vol. 35, no. 5, pp. 4026-4038, May 2025, doi: 10.1109/TCSVT.2024.3523359.
>
>
>  ---
>  > (Q4) The videos provided in the supplementary material are all shorter than 10 seconds. How does AUHead perform on audio clips longer than 10 seconds? Could the authors provide corresponding results?
>
> Thank you for the reviewer’s question. **AUHead supports generating videos longer than 10 seconds.** We provide long-sequence evidence in Figure 7 of the updated supplementary material. This figure presents frame sequences generated from 10 seconds unseen audio clips and target faces. To verify generalization, Figure 7 includes diverse target images such as line-art sketches, oil-painting portraits, and real faces. Each row is driven by a different long audio clip, and the displayed frames are randomly sampled from each second of the generated sequence. These results show that AUHead remains stable, expressive, and consistent over extended durations.

---

> ### Author Response · Authors · 2025-11-25
> **Gentle Reminder Regarding Our Previous Response**
>
> Dear Reviewer U6Er:
>
> Thanks a lot for your efforts in reviewing this paper! We tried our best to address the mentioned concerns and have provided a detailed response. We authors want to confirm whether there are unclear explanations and descriptions here. We could further clarify them.
>
> Thanks!
>
> Authors

---

### Official Review · Reviewer_SgZG · 2025-10-31

**Soundness:** 3
**Presentation:** 3
**Contribution:** 3
**Rating:** 6
**Confidence:** 5

**Summary:**

This paper presents AUHead, a two-stage framework for generating realistic and emotionally expressive talking head videos conditioned on speech and a reference image. The key idea is to employ Facial Action Units as an interpretable intermediate representation bridging audio and visual modalities, addressing the difficulty of producing fine-grained emotional expressions in prior works.
In Stage 1 (Understanding), an Audio Language Model (Audio-Qwen-Chat) extracts emotion aware AU sequences from speech through two steps: (1) a spatio-temporal AU tokenization strategy that compresses dense AU vectors, and (2) an emotion-then-AU Chain-of-Thought process that first predicts emotion categories before generating detailed AU sequences.
In Stage 2 (Controllable Generation), an AU-guided diffusion model synthesizes videos via three modules: (1) AU Representation, which upsamples and projects AU sequences into structured 2D facial maps (LMK or RoM) to ensure spatial realism; (2) Context-Aware AU Embedding, using temporal convolution to maintain motion consistency; and (3) AU Vision Interaction, which introduces AU-conditioned cross-attention and a classifier-free AU guidance mechanism for flexible control between expression fidelity and visual quality.

**Strengths:**

1. The supplementary visualizations are exceptionally clear and persuasive, providing strong qualitative evidence of the method’s superiority.

2. The two-stage pipeline is well justified, and adopting AUs as the intermediate feature is a sensible and effective choice.

3. As I know, this is the first work to leverage a large audio-language model to predict AU sequences from speech, enabling disentangled emotion-aware representations for talking-head generation.

4. Stage 2 effectively maps AUs to 2D spatial priors (landmark or mesh) and integrates temporal AU embeddings with AU–vision cross-attention adapters.

5. This paper is well-orgnized and well written.

**Weaknesses:**

1. The AU regression in Stage 1 achieves an MAE of around 0.2, which is comparable to inter-annotator variability. However, the paper does not explicitly discuss how this residual AU prediction error might influence the final video generation quality. This limitation is acceptable, but a short analysis would further clarify the robustness of the pipeline.

2. The AU sequences are downsampled to 5 fps to fit within the ALM’s context window, which may constrain the modeling of subtle, high-frequency facial movements.

3. The user study is small in scale, this is not sufficiently rigorous for strong perceptual claims and risks bias.

4. Some implementation details are briefly described and could be clarified. It would be useful to explain how the AU sequences are converted into 2D facial representations (landmarks and RoM).

**Questions:**

1. Does the chosen set of 24 AUs cover all the major emotional expressions present in the MEAD and CREMA datasets?

2. In the spatial-temporal AU tokenization, the AU sequences are downsampled to 5 fps and encoded as (index, intensity) pairs. Could the authors explain how this 5 fps rate was determined and whether alternative rates were considered?

3. Please provide more technical details on the AU to 2D representation mapping (LMK and RoM).

4. The user study involves a small number of raters. Could the authors elaborate on the selection process and whether they plan to expand this evaluation in future work?

---

> ### Author Response · Authors · 2025-11-21
> **Response to Weaknesses 1–4**
>
> We sincerely thank the reviewer for the thorough evaluation and helpful comments. Your feedback is highly valuable to us, and we respond to each weakness and question point-by-point in the following.
>
> ---
>
> > (W1)  The AU regression in Stage 1 achieves an MAE of around 0.2, which is comparable to inter-annotator variability. However, the paper does not explicitly discuss how this residual AU prediction error might influence the final video generation quality. This limitation is acceptable, but a short analysis would further clarify the robustness of the pipeline.
>
> Thank you for the helpful comment. Our Stage-1 AU regressor achieves an MAE of around 0.2, which is close to inter-annotator variability. As shown in Figure 4 and Figure 11, we did not observe noticeable distortions or unstable motion caused by this level of AU error. This observation is also consistent with the quantitative results in Table 3 and with the user study in Section 4.3.1.
> The remaining AU prediction errors mainly appear in subtle expression changes and are not systematic. To reduce their influence, we design two simple but effective mechanisms. First, the AU-Vision Interaction module conditions the generator on the AU sequence together with visual features, which makes the model rely less on small AU fluctuations. Second, the temporal window applies mild smoothing across adjacent frames, which limits frame-level jitter without oversmoothing the motion. In addition, the AU-CFG provides a stable way to adjust the guidance strength, helping the system maintain a good balance between AU control and visual quality. Together, these components make the pipeline robust to the small residual AU errors from Stage 1.
>
> ---
>
> > (W2) The AU sequences are downsampled to 5 fps to fit within the ALM’s context window, which may constrain the modeling of subtle, high-frequency facial movements.
>
> Please see the response to the following Q2.
>
> ---
>
>
> > (W3) The user study is small in scale, this is not sufficiently rigorous for strong perceptual claims and risks bias.
>
> Please see the response to the following Q4.
>
> ---
>
> > (W4) Some implementation details are briefly described and could be clarified. It would be useful to explain how the AU sequences are converted into 2D facial representations (landmarks and RoM).
>
> Please see the response to the following Q3.

---

> > ### Author Response · Authors · 2025-11-21
> > **Response to Question 1-4**
> >
> > > (Q1) Does the chosen set of 24 AUs cover all the major emotional expressions present in the MEAD and CREMA datasets?
> >
> > Thank you for the helpful question. The 24 AUs defined in FEAFA follow the long-established FACS [1,2] standard, whose definitions have been extensively validated and widely used in both facial analysis and facial animation research. It covers the major facial regions relevant to emotional expression, including the brows, eyes, nose, and mouth. These units include the core actions commonly used to represent emotion. As shown in Figure 4 and Figure 11, in fact, we did not observe expression patterns that required additional AUs.
> >
> >
> > [1]. Ekman P, Friesen W V. Facial action coding system[J]. Environmental Psychology & Nonverbal Behavior, 1978.
> >
> > [2]. Gan W, Xue J, Lu K, et al. Feafa+: an extended well-annotated dataset for facial expression analysis and 3d facial animation[C]//Fourteenth International Conference on Digital Image Processing (ICDIP 2022). SPIE, 2022, 12342: 307-316.
> >
> > ---
> >
> > > (Q2) In the spatial-temporal AU tokenization, the AU sequences are downsampled to 5 fps and encoded as (index, intensity) pairs. Could the authors explain how this 5 fps rate was determined and whether alternative rates were considered?
> >
> > Thank you for the reviewer’s helpful question. The 5 fps rate is mainly determined by the intrinsic redundancy of emotion cues in video, and the context window limit of current ALM. At the original 25 fps, a 4-second clip induces about13k AU tokens, which exceeds the typical 6k-token window and leads to unstable generation. Downsampling is therefore necessary. Among feasible rates, 5 fps offers a practical balance: it keeps the sequence short enough for reliable ALM processing while **preserving the key phases of facial actions (onset, peak, offset)**.
> >
> > In practice, we did not observe loss of perceptual smoothness at this rate. Qualitative results (Section 4.3: Qualitative Comparison) and the user study (Section 4.3.1: User Study) show that the generated expressions remain stable and natural. We agree that higher rates could capture finer details, and we plan to explore them as larger-context ALMs become available.
> >
> > ---
> >
> > > (Q3) Please provide more technical details on the AU to 2D representation mapping (LMK and RoM).
> >
> > Thanks for your suggestions. Specifically, we follow the Landmark (LMK) and Rendering-of-Mesh (RoM) pipelines introduced in BEAT[1] and EMAGE[2] to use a differentiable FLAME [3] model to convert facial control parameters into 3D geometry.  For our implementation,
> > 1) each 24-dimensional AU vector is first mapped to FLAME expression and jaw parameters, and thenuse the same ARKit-style mapping, where a learned linear transformation projects AU intensities into the FLAME parameter space. This allows the AU sequence to drive the corresponding 3D muscle movements in a stable and interpretable way.
> > 2) FLAME then produces a 3D mesh and 3D landmarks. It project them onto the image plane with a fixed camera model. The projected landmarks form the 2D LMK representation, and the rendered mesh forms the dense RoM image.
> >
> > [1] Haiyang Liu, Zihao Zhu, Naoya Iwamoto, Yichen Peng, Zhengqing Li, You Zhou, Elif Bozkurt, and Bo Zheng. Beat: A large-scale semantic and emotional multi-modal dataset for conversational gestures synthesis. arXiv preprint arXiv:2203.05297, 2022
> >
> > [2] Haiyang Liu, Zihao Zhu, Giorgio Becherini, Yichen Peng, Mingyang Su, You Zhou, Xuefei Zhe, Naoya Iwamoto, Bo Zheng, and Michael J Black. Emage: Towards unified holistic co-speech gesture generation via expressive masked audio gesture modeling. In CVPR, pp. 1144–1154, 2024.
> >
> > [3] Li T, Bolkart T, Black M J, et al. Learning a model of facial shape and expression from 4D scans[J]. ACM Trans. Graph., 2017, 36(6): 194:1-194:17.
> >
> > ---
> >
> > > (Q4) The user study involves a small number of raters. Could the authors elaborate on the selection process and whether they plan to expand this evaluation in future work?
> >
> > Thank you for the raising this concern.  To address it , we expanded the user study by recruiting **25 independent volunteers**. This larger and more diverse group reduces correlation risk and improves the reliability of the perceptual evaluation.
> > The expanded study follows the same blind-testing protocol, as detailed in Sec.4.3.1: User Study.  The results remain consistent with our initial findings: AUHead is preferred across all four aspects. Several raters also noted that the **audio–lip synchronization appears natural**. The updated results are reported in Table 4.
> >
> > | User Preference      | HalloV2 |   AUHead   |  Same  |
> > | -------------------- | :-----: | :--------: | :----: |
> > | Emotional Expression |  18.88% | **64.63%** | 16.49% |
> > | Video Quality        |  21.28% | **63.63%** | 15.09% |
> > | Audio-Lip Sync       |  13.75% | **71.00%** | 15.25% |
> > | Overall Performance  |  16.13% | **67.75%** | 16.12% |

---

> > > ### Comment · Reviewer_SgZG · 2025-11-22
> > >
> > > Thank you for the detailed clarifications and the expanded experiments. The additional explanations further strengthen the paper’s technical soundness, and the enlarged user study meaningfully improves the reliability of the perceptual evaluation. These revisions address my concerns, and I am comfortable raising my score to 8.

---

> > > > ### Author Response · Authors · 2025-11-25
> > > > **Thank You for the Helpful Feedback**
> > > >
> > > > Thank you very much for your thoughtful follow-up and for taking the time to reconsider the evaluation. We sincerely appreciate your constructive comments, which helped us clarify important technical points and improve both the experiments and the user study. Your feedback was truly valuable for strengthening the paper, and we are grateful for your supportive assessment.

---

> > > > > ### Author Response · Authors · 2025-11-29
> > > > > **Appreciation for Raising the Score to 8**
> > > > >
> > > > > Thank you once again for your kind follow-up and for your generous reassessment. **We sincerely appreciate your decision to raise the score to 8**, as well as the time and care you invested in reviewing our work.

---

### Meta-Review · Area_Chair_13PT · 2025-12-28

**Summary:**

This paper received 6 reviews. Initially, the review ratings are very diverse. During the short discussions, four reviewers either maintained the positive ratings or upgraded the rating: 3 rating 8 and 1 rating 6. One review from kSUe was overly harsh: the rating is 0. One rating from U6Er was 2. The author feedback to Reviewer U6Er  handled most comments.

Overall the reviews were positive. Please check more details in Section Reviewer Concerns.

**Reviewer Concerns:**

Reviewer SgZG

There were no critical comments; instead, nearly all feedback focused on clarifying descriptions, implementation details, and future work

Reviewer U6Er

One comment is about some audio-driven talking-head generation methods are not discussed or compared.
To the AC, the author feedback was convincing. The settings are not the same, e.g., some methods handle the problem of audio-driven Video2Video. The paper addresses the problem of audio-driven Image2Video.
The second comment is about the scale of the evaluation dataset. The author feedback was convincing: the dataset is standard in this area and has already been adopted by previous art.
The third coment is about more comparison with other methods are required. The authors provided more comparison. The feedback was almost fine, but the AC had a question: what's the difference from the settings of Hallo 3 and Hallo 4? They should be the same.
Other questions were minor.

Reviewer xY8X

The initial rating is 6. The reivewer provided a few comments about long video generation, the advantage of using AU compared to other intermediate representations, the loss of subtle details in facial expressions.

Regarding long video generation, the author feedback, "These results indicate that AUHead maintains identity consistency, expression stability, and smooth temporal transitions over extended durations.", is not quite convincing. And nowadays, 10 seconds is not long. However, the AC thought though this paper did not handle long video generation, the contribution was fine as an ICLR paper.

Other auther feedback are generally OK.

Reviewer kSUe

The comments were not particularly instructive, and they tended to be overly harsh. The AC will pay less attention to the comments. And The author responses were convincing.
(Initially, the review was flagged as very short by the original AC)

Reviewer rPph

Initially, the reviewer provided some comments, mainly about if using ALM is necessary, if AU sequence prediction is equivalent to "speech understanding", if the "AU-based disentanglement guidance strategy" proposed in the paper is original. The authors provided rebuttal. The reviewer had one more round discussions with the authors. Finally, the reviewer was convinced. The AC agreed with the reviewer.

Reviewer xaec

There were a few minor comments about user study and clafification on why ALM-based AU prediction is preferred over simpler audio–AU regression networks. The reviewer was convinced by the author rebuttal and promised to maintain the positive rating 8.

**Reviewer Scores:**

Reviewer SgZG

The reviewer already provided further comments: "These revisions address my concerns, and I am comfortable raising my score to 8.".

Reviewer U6Er

If the AC was this reviewer, most likely, the rating would be upgraded from 2.

Reviewer xY8X

The AC thought that most likely the review would maintain the rating 6.

Reviewer rPph

The reviewer already provided further comments: "I believe the revised paper has been greatly strengthened and now meets the acceptance criteria of ICLR. I will increase the score to 8"

Reviewer xaec

The reviewer was convinced by the author rebuttal and promised to maintain the positive rating 8.

---

### Decision · Program_Chairs · 2026-01-26

Accept (Poster)